# Zebra-CoT: A Dataset for Interleaved Vision-Language Reasoning

**Ang Li**[♠◇⋆]    **Charles L. Wang**[♠⋆]    **Deqing Fu**[♠⋆]    **Kaiyu Yue**[♡⋆]
**Zikui Cai**[♡⋆]    **Wang Bill Zhu**[♠⋆]    **Ollie Liu**[♠⋆]    **Peng Guo**[◇⋆]
**Willie Neiswanger**[♠]    **Furong Huang**[♡]    **Tom Goldstein**[♡]    **Micah Goldblum**[♠]

[♠]Columbia University    [◇]New York University    [♠]University of Southern California
[♡]University of Maryland
[⋆]Equal contribution

## Abstract

Humans often rely on visual aids, such as diagrams or sketches, when tackling complex problems. Teaching multimodal models to adopt similar strategies, a process known as Visual Chain of Thought (visual CoT), is much more difficult. The main challenges are: (1) weak performance of off-the-shelf visual CoT, which hinders reinforcement learning, and (2) the lack of high-quality visual CoT training data. We introduce **Zebra-CoT**, a diverse large-scale interleaved text–image reasoning dataset with 182,384 reasoning traces across 18 domains with over 50 distinct tasks. This dataset is specifically designed to train models to natively perform visual CoT. We emphasize four categories of tasks where sketching or visual reasoning is especially natural, spanning (a) *scientific questions* such as geometry, physics, and algorithms; (b) *2D visual reasoning tasks* like visual search and jigsaw puzzles; (c) *3D reasoning tasks* including 3D multi-hop inference, embodied and robot planning; and (d) *visual logic problems and strategic games* like chess. Fine-tuning Anole-7B model on Zebra-CoT yields a +12% improvement in our test-set accuracy and up to +13% performance gains on standard VLM benchmarks. Similarly, fine-tuning Bagel-7B produces models capable of generating high-quality interleaved visual reasoning chains, underscoring Zebra-CoT's effectiveness in advancing multimodal reasoning.

🦓 **Datasets**: multimodal-reasoning-lab/Zebra-CoT
🦎 **Anole-Zebra-CoT Model**: multimodal-reasoning-lab/Anole-Zebra-CoT
🥯 **Bagel-Zebra-CoT Model**: multimodal-reasoning-lab/Bagel-Zebra-CoT
🐙 **GitHub Repository**: github.com/multimodal-reasoning-lab/Bagel-Zebra-CoT

## 1 Introduction

Human cognition naturally integrates multimodal thought processes when solving complex problems. For example, a high school student sketches diagrams to solve geometry or physics problems, an engineer creates diagrams to design and debug workflows, and a data scientist generates plots to better understand data. These visual aids are central to effective problem solving. While recent vision-language models (VLMs) have shown strong performance on multimodal tasks like visual question answering, their reasoning traces remain predominantly textual. Enabling models to explicitly reason in the visual space, Visual Chain of Thought (visual CoT), remains a fundamental open challenge. Unlocking visual CoT may improve reasoning performance in domains where visual intuition is relevant and may make the reasoning patterns expressed by models more interpretable to humans.

Recent advances in frontier multimodal models (Team et al., 2023; Hurst et al., 2024; Bai et al., 2025; OpenAI, 2025a; Team, 2024; Chern et al., 2024; Sun et al., 2024; Deng et al., 2025) have made visual CoT feasible primarily through agentic pipelines that leverage external tools (*e.g.*, Python functions, or expert vision models) for visual programming (Surís et al., 2023), such as generating sketches for geometry, algorithms, and spatial reasoning tasks (Hu et al., 2024; OpenAI, 2025b), or bounding boxes for fine-grained visual tasks (Shao et al., 2024a; Wu and Xie, 2024; Zheng et al.,

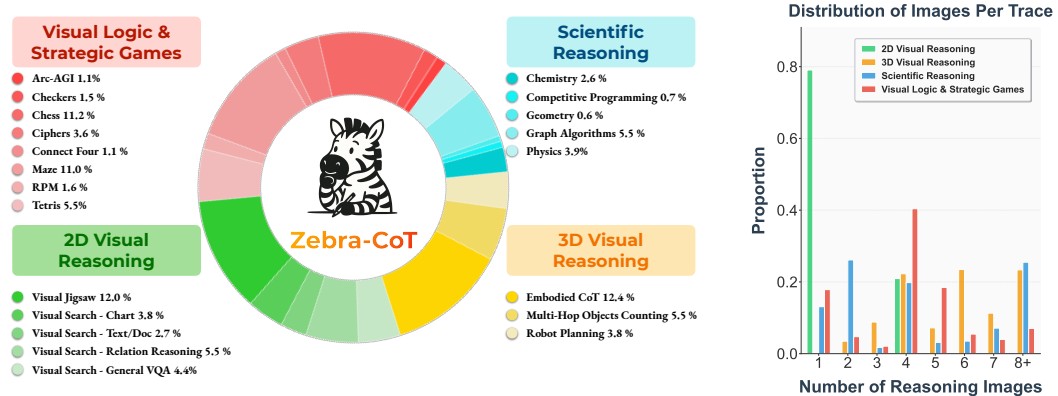

Figure 1: We curate a large-scale multimodal dataset by sourcing and cleaning raw traces from real-world domains, and generating synthetic examples using templated reasoning filled in by VLMs. ZEBRA-COT comprises 4 major categories and 18 subcategories, encompassing over **182K** instances in total. A detailed breakdown of the data statistics appears in Table 3.

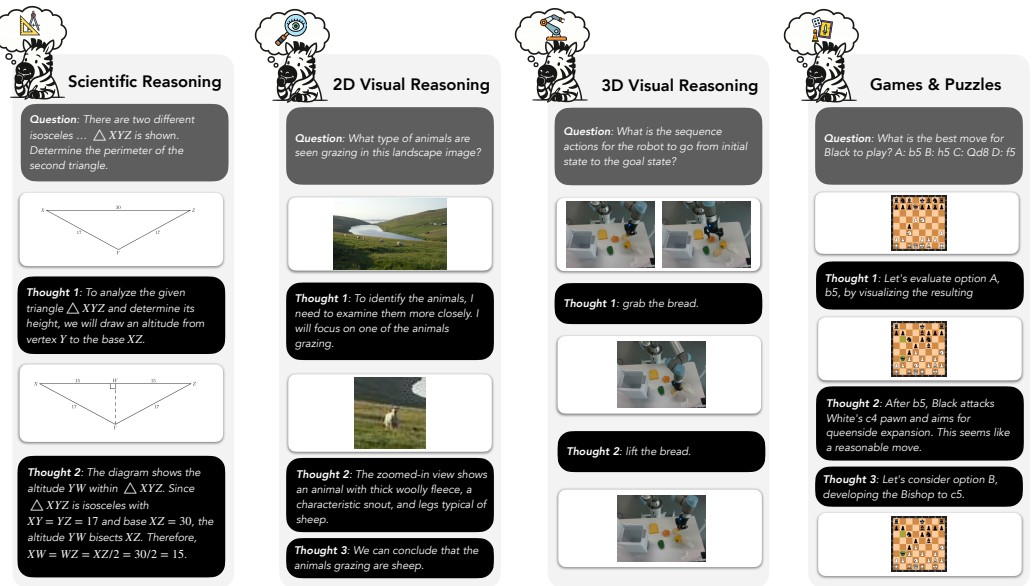

Figure 2: Visual CoT helps answer complex visual reasoning questions, as illustrated by examples from ZEBRA-COT.

2025). An emerging possibility is innate visual reasoning, where models directly generate explicit visual tokens during their thinking process (Li et al., 2025; Chern et al., 2025; Xu et al., 2025b). However, current VLMs with interleaved text and image generation capabilities (Team, 2024; Chern et al., 2024) either fail to generate useful visual aids for reasoning or are not inherently trained for such multimodal generation during the reasoning process (Deng et al., 2025), making reinforcement learning approaches to reasoning infeasible. Li et al. (2025) demonstrate visual CoT in synthetic mazes by training specialist models, but we remain far from foundation models capable of general high-quality visual CoT, largely due to the lack of large-scale diverse interleaved text and image reasoning training datasets.

To support the development of next generation vision language models that can explicitly reason with both text and visual modalities, we present **ZEBRA-COT**, a high quality dataset of interleaved text and image reasoning traces. Our dataset covers four main categories: scientific questions, 2D visual reasoning, 3D visual reasoning, and visual logic and strategic games, each containing multiple

| Dataset | Primary Task | CoT Modality | Suitability for visual CoT Training |
|---|---|---|---|
| GQA | Compositional visual QA | Text | No visual CoT |
| ScienceQA | Multimodal science QA | Text | No visual CoT |
| CLEVR | Synthetic compositional visual QA | Text | No visual CoT |
| VCR | Visual commonsense QA with rationale | Text | No visual CoT |
| VideoCoT | Video QA | Text | No visual CoT |
| EgoCOT | Embodied planning | Text | No visual CoT |
| LLaVA-CoT | Multimodal reasoning QA | Text | No visual CoT |
| MAmmoTH-VL | Large scale multimodal instruction tuning | Text | No visual CoT |
| MM-Verify | Multimodal reasoning with verification | Text | No visual CoT |
| R1-Onevision | A SFT and RL multimodal reasoning dataset | Text | No visual CoT |
| Visual CoT | Visual-search QA with bbox CoT | Image, Text | Limited to visual search tasks |
| MM-PhyQA | Physics visual CoT | Image, Text | Physics data only, not open sourced |
| CoT VLA | Robotics visual CoT | Image, Action | No text reasoning |
| OmniCorpus | 10 B-level interleaved corpus | None | Noisy pretraining data without CoT |
| MINT-1T | 1 T-token web-scale interleaved data | None | Noisy pretraining data without CoT |
| **ZEBRA-COT** | Diverse and high quality visual CoT | Image, Text | Diverse interleaved vision–language CoT |

Table 1: ZEBRA-COT introduces a broader set of high quality visual CoT traces compared with prior datasets and pipelines.

subdomains and task types, as exemplified in Figure 2. To the best of our knowledge, ZEBRA-COT is the first dataset to provide diverse and logically coherent multimodal reasoning traces across such a wide range of domains. Unlike prior large-scale interleaved datasets that are primarily composed of web-scraped image-text pairs with weak semantic alignment and no explicit reasoning structure (Li et al., 2024b; Awadalla et al., 2024; Zhu et al., 2023), ZEBRA-COT is carefully curated as a training resource in the spirit of high-quality text-based reasoning datasets. At the same time, compared to the only existing open-source interleaved text visual reasoning dataset we are aware of, VISUAL-COT (Shao et al., 2024a), which focuses on a single task of visual search, ZEBRA-COT introduces a much broader and more diverse set of tasks with richer reasoning trajectories. We provide a detailed comparison with other datasets below in Table 1.

**Our contributions are summarized as follows:**

1. We release ZEBRA-COT, a high quality and diverse dataset with interleaved text and visual CoT that contains 182,384 samples for training models to natively perform visual CoT for problem solving. Details regarding the dataset are shown in Section 3

2. We evaluate three frontier LLMs, including GPT-5, Claude Sonnet 4, and Gemini 2.5 Pro, on the tasks in ZEBRA-COT in Section 4. Despite their advanced multimodal reasoning capabilities, these models perform poorly on those challenging tasks, with an average of 31.51%. Moreover, to demonstrate the effectiveness and value of visual CoT, we construct a scaffolding experiment that provides the first one or two multimodal CoT steps in context. Accuracy rises to 47.99% after one step (**+16.48** pts) and 56.70% after two steps (**+25.19** pts) overall, with gains of up to **+43.77** pts in specific domains. These findings highlight the challenging nature of our dataset, the quality of our reasoning traces, and the value of visual CoT.

3. After fine-tuning ANOLE-7B (Chern et al., 2024) on our training set, we improved the accuracy on our in-distribution test set from 4.2% to 16.9%. When evaluating the resulting model on benchmarks requiring visual reasoning, our ANOLE-ZEBRA-COT-7B model achieves an average improvement of **4.9%** across seven challenging datasets, with a maximum gain of **13.1%** on a visual logic benchmark, as shown in Table 2.

4. We fine-tune BAGEL-7B (Deng et al., 2025), a high-quality multimodal model that cannot natively generate interleaved text and images on our dataset. After fine-tuning, the model is able to inherently generate high-quality visual CoT during its own reasoning process, making it well-suited for future RL training, as shown qualitatively in the examples in Figure 4 and Appendix B.

## 2 RELATED WORK

**Visual chain of thought.** The community has predominantly been tackling visual CoT by using visual programming to generate images (Surís et al., 2023; Zhang et al., 2023; Mitra et al., 2024; Yang* et al., 2023; Wu and Xie, 2024; Hu et al., 2024; Menon et al., 2024; OpenAI, 2025b; Zheng et al., 2025). In particular, VISUAL SKETCHPAD (Hu et al., 2024) presents the most versatile open-source visual reasoning agents among existing works, handling a wide range of tasks. Another line of work explores model-generated images: for example, Rose et al. (2023) uses a diffusion model to bridge gaps in storytelling, and Chern et al. (2025) generates intermediate images to improve image generation tasks; Zhao et al. (2025) generates intermediate images as subgoal predictions and derives actions based on them for robotic planning; Li et al. (2025) and Xu et al. (2025b) explore spatial reasoning tasks like mazes by visualizing each temporal step. However, these model-generated image approaches are mostly specialists, and developments are still primitive compared to visual programming methods that leverage external tools.

**Visual reasoning datasets.** Many multimodal visual reasoning datasets have been proposed, such as GQA (Hudson and Manning, 2019), SCIENCEQA (Lu et al., 2022), VIDEOCOT (Wang et al., 2024c), EGOCOT (Mu et al., 2023), LLAVA-COT (Xu et al., 2024), MAMMOTH-VL (Guo et al., 2024), MM-VERIFY (Sun et al., 2025), R1-ONEVISION (Yang et al., 2025), CLEVR (Johnson et al., 2017), VCR (Zellers et al., 2019), although most focus on multi-modality only in the input question, leaving the reasoning traces purely textual. Among them, VISUAL-COT (Shao et al., 2024a) stands out as the only open-source dataset featuring interleaved text and image reasoning. MM-PHYQA (Anand et al., 2024) on the other hand, introduces a paradigm for incorporating images into the reasoning process for physics problems, though the dataset is not publicly available. Several vision-centric benchmarks (Fu et al., 2024b; Hao et al., 2025a) present diverse and challenging tasks, but they lack annotated reasoning traces.

**Interleaved text and image datasets.** Large-scale corpora with interleaved text and images have become essential for pretraining VLMs with reasoning capabilities (Alayrac et al., 2022; Chen and Wang, 2022; Sun et al., 2024; Wang et al., 2024b; Hurst et al., 2024; Li et al., 2024a; Bai et al., 2025; Team et al., 2025). However, in most existing interleaved text and image datasets MULTIMODAL C4 (Zhu et al., 2023), OBELICS (Laurençon et al., 2023), OMNICORPUS (Li et al., 2024b), images are primarily used for recognition, captioning, or as supplementary context in text-based reasoning, rather than serving as explicit visual aids that contribute meaningfully to the reasoning process. While MINT-1T (Awadalla et al., 2024) includes some scientific content from arXiv where images may aid reasoning, both the text traces and visual content are often noisy and not well-suited for post-training or fine-grained reasoning tasks. Instead, our ZEBRA-COT introduces a broader and higher-quality set of visual CoT examples, enabling effective training for visual reasoning.

## 3 DATA CURATION DETAILS AND COMPOSITIONS

### 3.1 CURATING A DIVERSE AND HIGH QUALITY INTERLEAVED VISION AND LANGUAGE REASONING DATASET

A key challenge in training multimodal generation models to output visual CoT natively is the lack of datasets with strong logical coherence between text and visual modalities, and diverse categories of such visual CoT. Existing interleaved datasets often fail to provide clear, meaningful connections that demonstrate when and why visual reasoning is necessary for problem-solving, while current visual CoT datasets are confined to a few domains, limiting the model's ability to learn generalizable visual CoT capabilities when faced with out-of-distribution problems.

To address these gaps, we developed a comprehensive data curation pipeline that bridges logical connections across modalities, as shown in Figure 5. For logical coherence across modalities, we leverage frontier vision-language models (Gemini-2.5 Pro) to enrich reasoning traces and ensure a clear logical flow between textual reasoning and visual aids. For diversity, we combine real-world problems from multiple domains (mathematics, physics, chemistry, coding, chess, visual question answering, robotics) with synthetic examples generated through computer programming, simulation, and graphic rendering. This pipeline enabled us to curate over 182 K high-quality interleaved text and visual reasoning traces spanning four major categories: scientific reasoning, 2D visual reasoning, 3D visual reasoning, and visual logic and strategic games. Unlike existing limited datasets

that focus primarily on visual search or spatial reasoning, our curated dataset provides the breadth and diversity necessary for training models that can generalize across domains. For details regarding our data curation pipeline, please refer to Appendix A.2. In the following sections, we provide a brief introduction to the tasks of each broad category. And for the details regarding the subcategory and domains, please refer to Appendices A.3 to A.6. For prompt templates, please refer to Appendix F.

## 3.2 SCIENTIFIC QUESTIONS

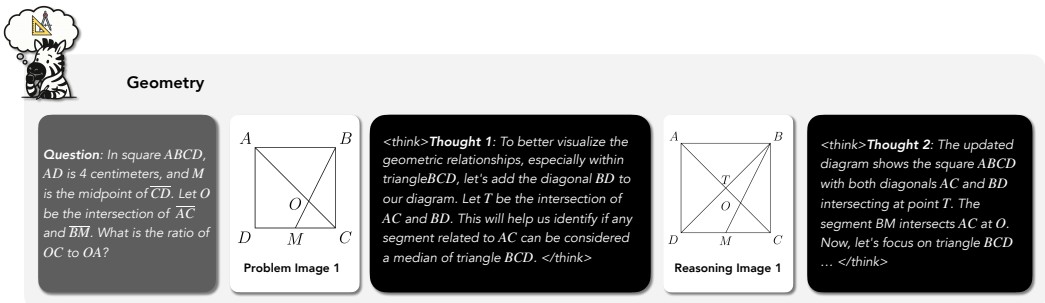

Visual reasoning is particularly valuable in STEM domains, as it enables the visualization of abstract concepts such as auxiliary lines, free-body diagrams, and sketches, which clarify ideas that are hard to describe in language and support step-by-step problem solving in ways that mirror human cognition. In ZEBRA-COT, this category spans subdomains including geometry, physics, chemistry, algorithmic problem solving, and graph problems. For geometry, physics, and chemistry, we leverage openly licensed datasets and textbooks, using Gemini-2.5 (Comanici et al., 2025) to denoise and parse them into clean, logically structured visual CoT. For graph problems, we employ computer programs to generate images and text templates, which are then diversified using Gemini-2.5. For algorithmic problems, we use a GPT-4.1 agent built upon Hu et al. (2024) to produce detailed traces for solving competitive programming tasks. For details regarding all tasks in this domain, see Appendix A.3.

## 3.3 2D VISUAL REASONING

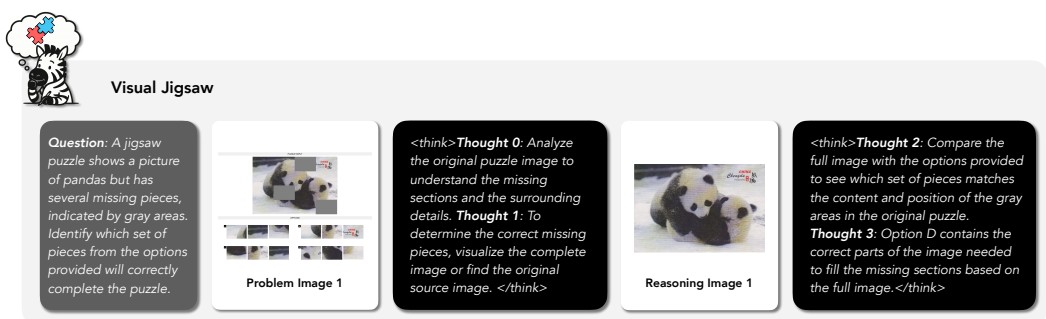

In 2D visual reasoning, visual aids support the manipulation and interpretation of 2D visual information, enabling tasks that involve spatial arrangement, pattern recognition, and fine-grained inspection. For this category, we include tasks such as visual search and visual jigsaw. For visual search, we adapt datasets from Shao et al. (2024a) and incorporate two types of visual aids: drawing bounding boxes and zooming into focal regions. We apply those visual CoT broadly across data categories, such as charts, documents, relations, and general VQA. For visual jigsaw tasks, we crop images from ImageNet (Deng et al., 2009) to create puzzles with a random number of missing pieces in diverse shapes. The visual CoT is either iteratively filling in the pieces or reconstructing the original image directly. Further details are provided in Appendix A.4.

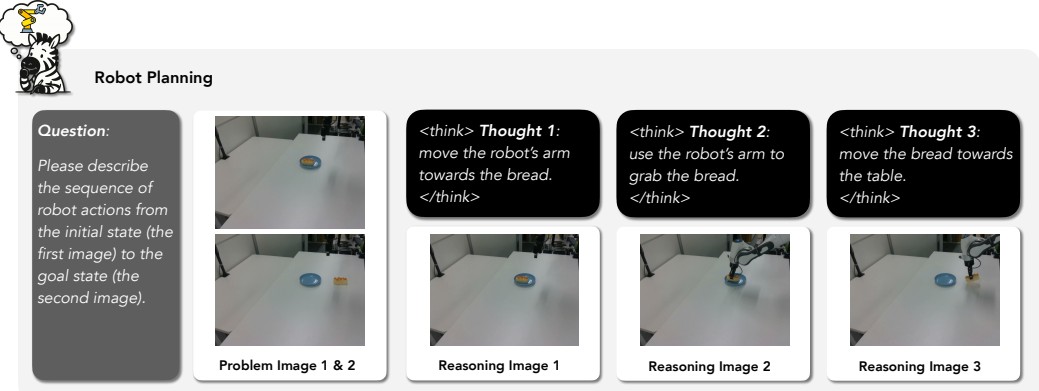

## 3.4 3D Visual Reasoning

For 3D visual reasoning tasks, we focus on two domains: (1) embodied reasoning and robotic planning in the physical world, and (2) understanding 3D transformations from different viewpoints. For the first domain, prior work has shown that generating visual predictions of the physical world and extracting inverse dynamics can improve performance in long-horizon decision-making in robotics (Zhao et al., 2025; Yang et al., 2024). To capture this, we reformulate the ALFRED benchmark (Shridhar et al., 2020) into an image goal-conditioned planning task in which models generate detailed step-by-step plans to transition from an initial state to a goal state. We also adapt RoboMIND (Wu et al., 2024) for real-world robot planning, where models receive initial and goal images, along with descriptions of robot embodiment, and must produce precise high-level action plans. For reasoning about 3D transformations, we design multi-hop object counting tasks inspired by CLEVR (Johnson et al., 2017), where scenes undergo sequential modifications, such as adding or removing objects, requiring models to visually reason through each transformation step. For details, see Appendix A.5

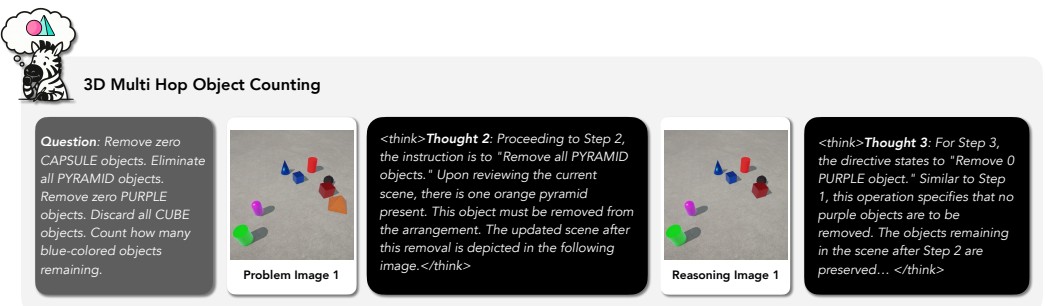

## 3.5 Visual Logic and Strategic Games

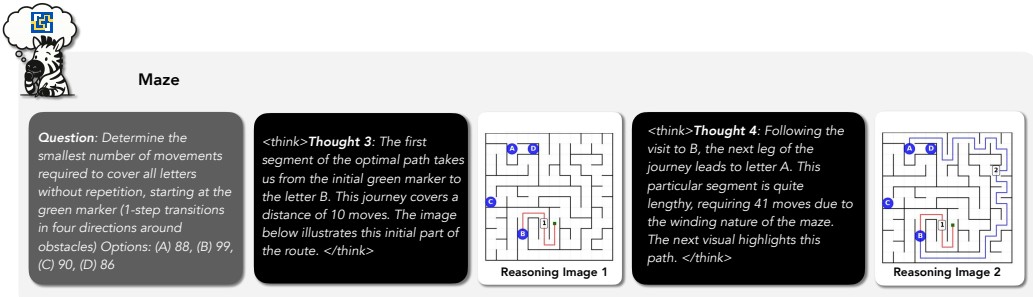

For visual logic puzzles (IQ matrices, Tetris, ciphers, ARC-AGI (Chollet et al., 2024)), previous VLMs tended to solve problems primarily using text reasoning. They first verbalize visual inputs

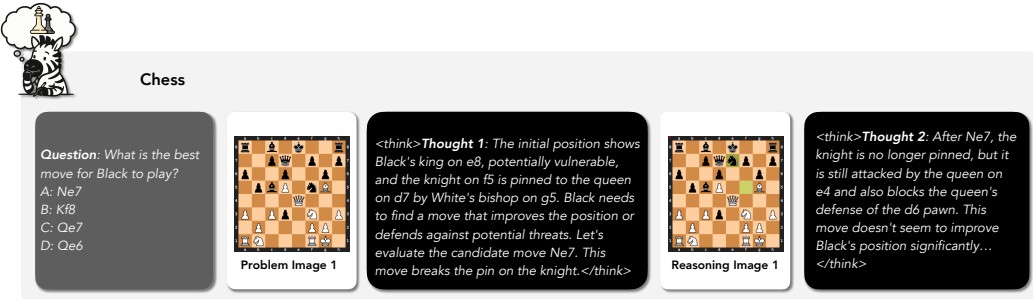

into text, which causes information loss and makes visually salient patterns, such as spatial relationships, difficult to capture. In contrast, humans solve these directly and efficiently via visual imagination and manipulation, even for babies who have not yet acquired language capabilities (Zhu et al., 2020). To bridge the gap, we construct visual CoT traces that include explicit intermediate visual transformations to encourage models to solve these problems visually. Similarly, for strategic games (chess, checkers, Connect Four), decision making typically involves searching and generating counterfactual rollouts. While LLMs can simulate this by symbolizing board states into text, much of the spatial structure is lost, and rollouts in text space are difficult for problems with large visual information. Thus, we render those search and simulation steps into images so that models trained on this data can perform long-horizon planning in the visual space inherently. Finally, we generate a diverse suite of maze tasks and visual CoT traces that require a combination of capabilities, including high-level symbolic search and low-level perception. For details of those tasks, see Appendix A.6.

## 4 ANALYSIS OF ZEBRA-COT AND THE VALUE OF VISUAL COT

Proprietary frontier models (GPT-5 (OpenAI, 2025c), Gemini-2.5 Pro (Comanici et al., 2025), Claude-4 Sonnet (Anthropic, 2025)) have achieved state-of-the-art performance on multimodal reasoning benchmarks. Despite their advanced multimodal capabilities, we show that they struggle significantly with the tasks in ZEBRA-COT. To explore these limitations and demonstrate the challenging nature of our dataset alongside the effectiveness of visual reasoning traces, we design a scaffolding experiment. Specifically, our dataset consists of structured reasoning chains: `<question>` → `<text-reasoning-1>` → `<visual-reasoning-1>` → `<text-reasoning-2>` → `<visual-reasoning-2>` → ... → `<answer>`.

In the zero-shot setting, we provide models only with the `<question>` (containing both image and text). For scaffolding experiments, we incrementally provide the first $k$ multimodal reasoning steps as context:

- **1MT** ($k = 1$): `<question>` + `<text-reasoning-1>` + `<visual-reasoning-1>`
- **2MT** ($k = 2$): `<question>` + `<text-reasoning-1>` + `<visual-reasoning-1>` + `<text-reasoning-2>` + `<visual-reasoning-2>`

Importantly, most tasks in ZEBRA-COT require various multimodal reasoning steps (which can involve as many as 20 images) to reach the final answer. By providing only the first two steps as scaffolding, we ensure that models must still perform substantial reasoning to solve the task. We can safely assume that the provided steps serve as guidance rather than revealing the solution. Since our dataset comprises diverse tasks, some of which extend beyond traditional QA formats (e.g., robotic planning and embodied CoT) that are not suitable for evaluation, we select the most challenging and representative examples for evaluation: graph questions for scientific reasoning, visual jigsaw for 2D spatial reasoning, multihop object counting for 3D reasoning, and maze/chess/tetris for visual logic and strategic games.

We plot the results for three evaluation settings across each task domain in Figure 3. We observe that frontier models achieve poor zero-shot performance: GPT-5 reaches 41.98% accuracy, while Claude-4 Sonnet and Gemini-2.5 Pro achieve only 27.61% and 24.93% respectively. However, with multimodal CoT scaffolding, we observe substantial improvements: average accuracy across the three models increases to 47.99% (**+16.48%**) with one reasoning step and 56.70% (**+25.19%**) with two steps.

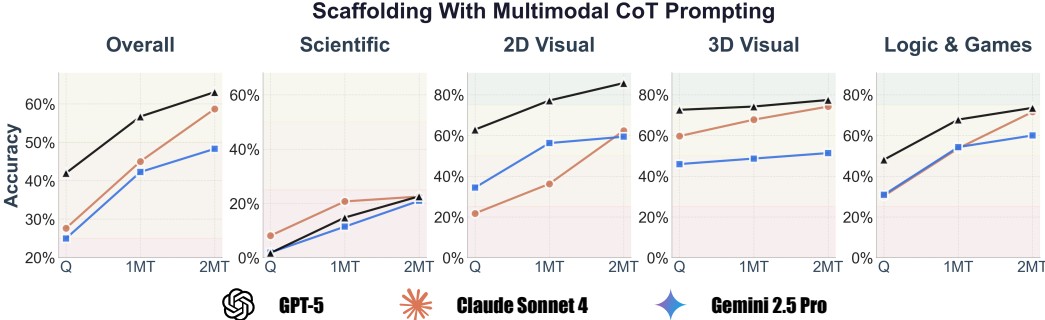

Figure 3: Scaffolding experiment with frontier models. **Q** represents zero-shot question-only evaluation, **1MT** denotes a question with the first multimodal reasoning step provided, and **2MT** indicates a question with the first two multimodal reasoning steps. We show that **even frontier models with the best multimodal reasoning capabilities perform poorly overall on tasks in ZEBRA-COT**. However, as we provide the first one or two multimodal steps to those models, the accuracy improves significantly.

Performance gains vary across task types, but we generally see an improvement trend. Maze tasks show the most dramatic improvements, which jump from 52.59% to 76.60% (+24.01%) and to 96.36% (+43.77%) on average, while challenging tasks such as graph reasoning improve from 3.92% to 22.03% (+18.11%) with two multimodal reasoning steps on average. Even tasks with higher baseline performance, such as multihop object counting (with an initial accuracy of 59.40%), benefit from visual CoT, eventually reaching 67.65% accuracy on average. Detailed statistics are shown in Table 8.

To isolate the contribution of visual reasoning aids from text CoT in our traces, we conduct an ablation task where we remove all visual aids from the reasoning traces and retain only the textual steps. We observe that text-only CoT yields substantially smaller performance gains compared to full visual CoT, and in some cases even degrades performance. This is expected: in our dataset, the visual and textual components are highly complementary. Many reasoning steps reference visual elements that, once removed, leave the text chain logically incomplete or incoherent. Model even requests for the missing visual aids that are referred in the text cot. These results indicate that the majority of the performance improvements stem from the visual reasoning steps, or the combined visual + text reasoning, rather than from textual CoT alone. The statistics for text only results are shown here:

## 5 TRAINING MODELS ON ZEBRA-COT

| Model | MathVision[*] | MathVista[*] | VisuLogic | EMMA | MMVP | Blink | Vstar |
|---|---|---|---|---|---|---|---|
| Anole with CoT prompting | 13.80 | 22.80 | 8.50 | 12.80 | 10.00 | 26.46 | 23.60 |
| **Anole–Zebra-CoT (Ours)** | **16.45** | **25.30** | **21.80** | **15.02** | **15.33** | **31.25** | **27.20** |

Table 2: Overall performance (%) across eight datasets for the base Anole model with chain-of-thought prompting vs. the same Anole model further trained on ZEBRA-COT. [*]We evaluate on the mini versions of MathVision and MathVista because interleaved generation is time consuming. A full breakdown of each evaluation set is presented in Appendix C.

**Anole-Zebra-CoT.** We fine-tune Anole (Chern et al., 2024) on our dataset, which builds on Chameleon (Team, 2024), using the codebase from Chern et al. (2025). We finetune the model fully end-to-end on a node with $8 \times$ H200 GPUs for 12 hours, with a learning rate of $1 \times 10^{-5}$, cosine decay, a batch size of 8, and a max token length of 12288. We train the model for 10k steps. To evaluate our trained model, we set the maximum generation length to 16384. After fine-tuning Anole on our ZEBRA-COT corpus, the accuracy increased from 4.2% to 16.9%, delivering a 4 times relative performance improvement and a 12% gain in accuracy.

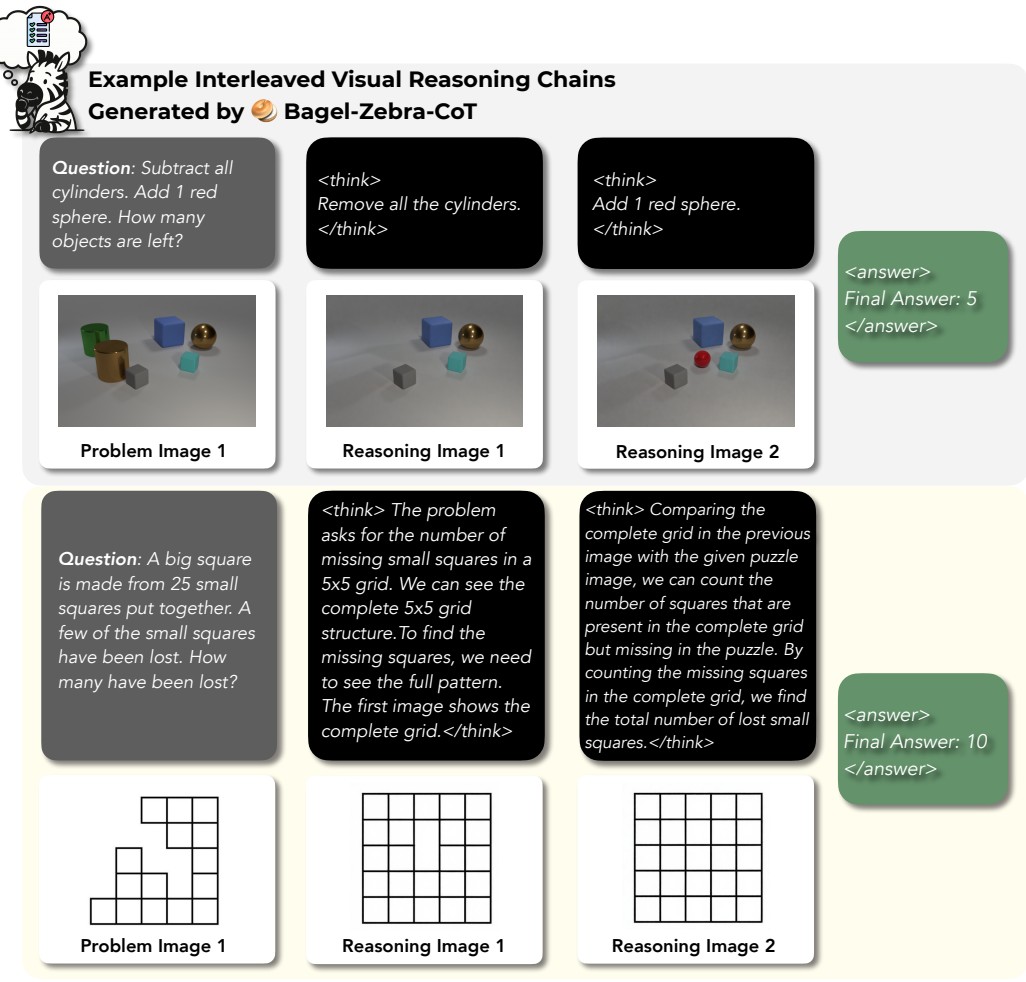

Figure 4: Example interleaved reasoning chains generated by Bagel-Zebra-CoT, a Bagel-7B model finetuned on ZEBRA-COT. These traces demonstrate ZEBRA-COT's for instilling intrinsic visual reasoning capability in complex multimodal models.

Furthermore, we evaluate seven challenging benchmarks that require visual reasoning, including `MathVision` (Wang et al., 2024a), `MathVista` (Lu et al., 2024), `VisuLogic` (Xu et al., 2025a), `EMMA` (Hao et al., 2025b), `MMVP` (Tong et al., 2024), `BLINK` (Fu et al., 2024b), and `Vstar` (Wang et al., 2023). All the evaluations are done using `VLMEvalKit` (Duan et al., 2024). To ensure a fair comparison, we use chain-of-thought prompting (Wei et al., 2022) when evaluating the base Anole model. As shown in Table 2, training with ZEBRA-COT significantly improves the Anole model across all benchmarks. Most notably, it could improve the Anole model's visual logical reasoning capabilities by 13.3 points.

**Bagel-Zebra-CoT.** To further test whether ZEBRA-COT can enhance a stronger backbone, we fine-tune the BAGEL-7B model (Deng et al., 2025) end-to-end on a node with $8\times$ H200 GPUs for 1,000 steps using packed sequences with 60,000 tokens, a learning rate of $2 \times 10^{-5}$, and cosine decay. We cap all images at a resolution of 512 on the minimum side, resulting in approximately $1,024+$ visual tokens per image. Because the original Bagel implementation cannot natively generate interleaved text–image outputs, we revise the training loop to include a loss term at the `<|vision_start|>` token, enabling seamless visual token generation. We additionally wrap text reasoning tokens between `<think>` and `</think>`, and the final answer within `<answer>` and `</answer>`. At inference time, when encountering `<im_end>`, we sample one additional token to check whether the next token is `<|vision_start|>`; if so, the model itself seamlessly switches to image gener-

ation mode to generate visual aids. The entire interleaved generation process only stops if the model generates the `<answer>` token.

We observe that our trained model can inherently generate visual CoT when solving problems, even on tasks outside its training distribution. This suggests its potential as a strong initialization for future reinforcement learning fine-tuning. In Figure 4, we include representative reasoning traces produced by the model. We further include more reasoning traces in Appendix B, as well as a model performance analysis in Appendix D

## 6 Conclusion & Future Directions

In this paper, we introduced ZEBRA-COT, a large-scale dataset of 182K interleaved text-image reasoning traces spanning 4 major categories across 18 domains with over 50 distinct tasks. Fine-tuning experiments demonstrate substantial improvements: Anole-7B achieves an average 4.9 % gain across seven challenging benchmarks, with up to 13.1% on visual logic tasks, while Bagel-7B learns to inherently generate visual aids during problem solving, a capability absent in the base model.

This work opens several exciting avenues for future research. Most immediately, models trained on ZEBRA-COT, particularly our Bagel variant that natively generates visual thoughts, provide strong initializations for reinforcement learning. Just as text-based reasoning models have benefited from RL fine-tuning to improve logical consistency and correctness, we envision similar gains for visual reasoning through RL with verifiable rewards (Shao et al., 2024b; Guo et al., 2025) or fine-grained rewards (Zeng et al., 2024; Fu et al., 2025).

We believe ZEBRA-COT represents a crucial step toward AI systems that think visually as naturally as humans sketch diagrams, generate graphs, and use spatial reasoning to solve complex problems. With our dataset and fine-tuned model, we hope to accelerate progress toward this goal.

## 7 LLM Usage Disclosure

We used LLM for two purposes. The first one is for improving grammar and wording when writing the paper. The second usage is synthetic data generation, where details can be found in Section 3 and Appendix A.2

## Acknowledgments

MG and AL were supported by a Research Award from the Columbia Center of AI Technology in collaboration with Amazon and by Google. KY, ZC, FH, and TG were supported by DARPA Transfer from Imprecise and Abstract Models to Autonomous Technologies (TIAMAT) 80321, National Science Foundation NSF-IIS-2147276 FAI, and DOD-AFOSR-Air Force Office of Scientific Research under award number FA9550-23-1-0048. WN was supported in part by the National Science Foundation under Grant No. CMMI-2427856. OL, DF, and BZ would like to thank Center for Advanced Research Computing (CARC) and NLP Group at USC for providing compute resources. In addition, Compute resources were generously provided by the NVIDIA Academic Grant program. Any opinions, findings, and conclusions or recommendations expressed in this material are those of the author(s) and do not necessarily reflect the views of the National Science Foundation.

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

# A    DATASET DETAILS

## A.1    DATA STATISTICS.

Here we show detailed statistics about ZEBRA-CoT's categories.

Table 3: Statistics of ZEBRA-CoT.

| General Category | Sub Category | Count | Percentage (%) |
|---|---|---|---|
| 2D Visual Reasoning | Visual Jigsaw | 21,899 | 12.0 |
| | Visual Search | 30,000 | 16.4 |
| | **Subtotal** | **51,899** | **28.5** |
| 3D Visual Reasoning | Embodied Cot | 22,666 | 12.4 |
| | Multi-Hop Objects Counting | 10,000 | 5.5 |
| | Robot Planning | 6,944 | 3.8 |
| | **Subtotal** | **39,610** | **21.7** |
| Scientific Reasoning | Chemistry | 4,666 | 2.6 |
| | Competitive Programming | 1,207 | 0.7 |
| | Geometry | 1,058 | 0.6 |
| | Graph Algorithms | 10,000 | 5.5 |
| | Physics | 7,090 | 3.9 |
| | **Subtotal** | **24,021** | **13.2** |
| Visual Logic Strategic Games | Arc-Agi | 2,000 | 1.1 |
| | Checkers | 2,753 | 1.5 |
| | Chess | 20,483 | 11.2 |
| | Ciphers | 6,589 | 3.6 |
| | Connect Four | 2,029 | 1.1 |
| | Maze | 20,000 | 11.0 |
| | RPM | 3,000 | 1.6 |
| | Tetris | 10,000 | 5.5 |
| | **Subtotal** | **66,854** | **36.7** |
| **Total** | | **182,384** | **100.0** |

## A.2 CURATING DIVERSE AND HIGH QUALITY VISUAL COT

**Bridging logical connections across modalities.** A key challenge in training multimodal generation models to output visual CoT natively is the lack of datasets with strong logical coherence between text and visual modalities, and diverse categories of such visual CoT. Existing interleaved datasets often fail to provide clear, meaningful connections that demonstrate when and why visual reasoning is necessary for problem-solving, while current visual CoT datasets are confined to a few domains, limiting the model's ability to learn generalizable visual CoT capabilities when faced with out-of-distribution problems.

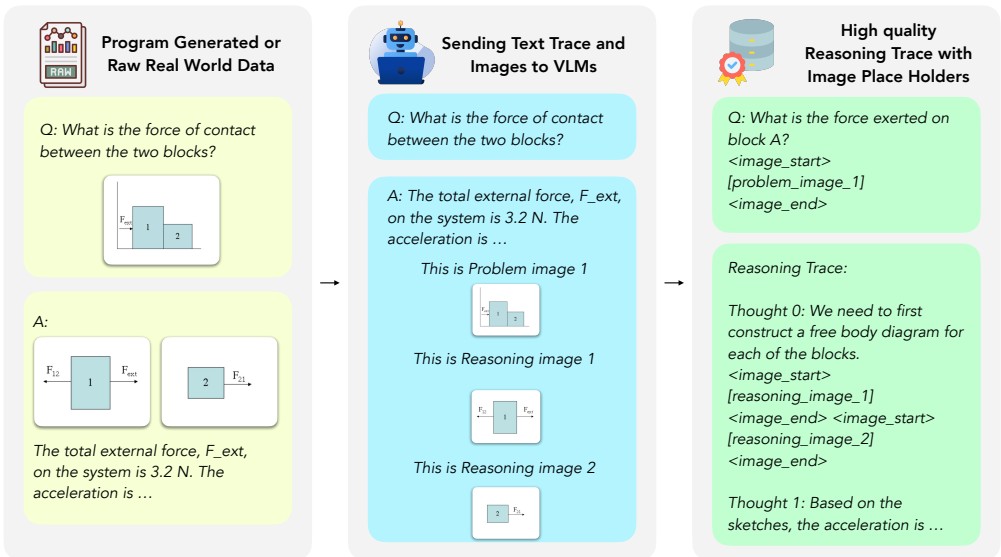

Figure 5: An overview of our data curation pipeline.

To address these requirements, we first source a diverse range of question types and domains. For real world data, we source high-quality problems from online resources such as math, physics, coding, and chess competition datasets. We then extract and clean the available raw reasoning traces that contain text and images. However, even from high quality sources, traces can still lack clear logical connections between modalities, as well as clear references to the images for automatic parsing into interleaved text and image data ready for training. For example, most geometry data uses reference labels such as "Figure $x$", which makes it hard to find the mapping between the actual image and the text reference. For synthetic data, we create our own examples by generating images or utilizing real images from online sources, then crafting corresponding reasoning templates. This procedure raises a clear issue, namely that we lack diversity and expressiveness in textual reasoning regarding templated data. For instance, in visual search tasks, it is crucial to elucidate the rationale behind drawing specific bounding boxes, and in chess, generating reflections and descriptions of move visualizations is key.

We address both of these issues using frontier VLMs (Gemini-2.5 and GPT-4.1) to fill in the template placeholders, enhance the reasoning traces, and complete the textual reasoning narrative. We feed both images and raw text reasoning traces into the language model and ask the language model to output pure text traces with image placeholders. We further filter out invalid cases, such as multiple image placeholders referring to the same image and unreferenced image placeholders, to ensure that the data can be automatically parsed into a training dataset.

**Broadening breadth and diversity of interleaved visual language reasoning dataset.** Furthermore, existing multimodal rationale datasets are also limited in their breadth. The only available datasets focus on either visual search (Wu and Xie, 2024; Shao et al., 2024a) or spatial reasoning, such as maze navigation (Li et al., 2025). Such limited datasets are unlikely to enable training visual reasoning models that can generalize across domains more broadly. Visual Sketchpad (Hu et al., 2024) offers a diverse range of VLM agents to tackle a wider variety of questions. Though Sketchpad offers a powerful and significant contribution to generating visual aids, the pipeline is

not designed for collecting post-training datasets. First, the reasoning traces generated by agentic pipelines often involve tool call errors and debug information, which degrade their quality. Second, the scalability and diversity of the dataset are fundamentally constrained by the limited number of agent tool designs and the high cost, as each reasoning trace may require many API calls. To tackle these issues, we curate a total of over **182K** high-quality interleaved text and visual reasoning traces, spanning four major categories: scientific reasoning, 2D visual reasoning, 3D visual reasoning, and visual logic and strategic games. We provide the details in the section below and example traces from our dataset.

### A.3 Scientific Questions

**Geometry.** Geometric understanding is a core ability for multimodal models to ground reasoning over complicated mathematical tasks. Many datasets have been proposed to evaluate mathematical capabilities, including geometry. The MATH dataset (Hendrycks et al., 2021) is widely used for evaluating the mathematical performance of LLMs. Although the MATH dataset includes numerous geometry competition problems, their geometric elements are provided as plotting code rather than rendered images (see Figure 6).

Here, we provide example code for geometry sketch generation.

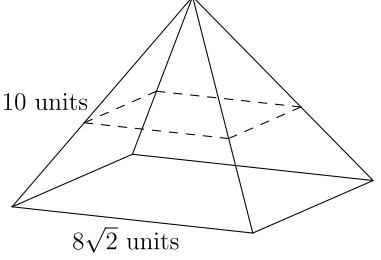

MATH/GEOMETRY/44

```
[asy]
import three;
size(2.5inch);
currentprojection =
orthographic(1/2,-1,1/4);
triple A = (0,0,6);
triple[] base = new triple[4];
base[0] = (-4, -4, 0);
base[1] = (4, -4, 0);
base[2] = (4, 4, 0);
base[3] = (-4, 4, 0);
triple[] mid = new triple[4];
for(int i=0; i < 4; ++i)
mid[i] = (.6*xpart(base[i]) +
.4*xpart(A), .6*ypart(base[i]) +
.4*ypart(A), .6*zpart(base[i]) +
.4*zpart(A));
for(int i=0; i < 4; ++i){
draw(A--base[I]);
draw(base[i]--base[(i+1)%4]);
draw(mid[i]--mid[(i+1)%4],
dashed);
}
label(``8√2 units",
base[0]--base[1]);
label(``10 units", base[0]--A,
2*W);
[/asy]
```

(a) Geometric Example in ZEBRA-CoT

(b) Geometric Example in MATH Dataset (Hendrycks et al., 2021)

Figure 6: Comparison of the same geometric figure in our ZEBRA-CoT dataset and the MATH dataset. Ours focus on multimodal reasoning and explicitly plot the geometry problem than using the text-only plotting codes.

In ZEBRA-CoT, we convert every piece of plotting code into figure renderings, producing both the problem diagram and its solution illustration to serve as an explicit visual reasoning chain for model training.

In total, we collect 1,061 samples from the MATH dataset's train split. Our data provides only rendered images for both the problem and solution reasoning chains, with no plotting code included. Solving these problems requires generating images to assist. The problems are not restricted to the `geometry` subcategory but also include some problems from `counting and probability`, `pre-algebra`, `pre-calculus`, etc.

**Physics.**    A variety of physics problems benefit from sketches, such as free body diagrams for force analysis, motion diagrams for kinematics, circuit diagrams for electricity, and ray diagrams in optics. We construct samples of classical mechanics problems programmatically. Problem instances are generated from parametric Python templates (e.g., Atwood machines, inclined planes, elastic collisions, pendulums), with physically plausible parameters sampled from predefined ranges. For each sample, we render free-body diagrams, kinematic visuals, and structured CoT traces capturing the full solution process.

We also leverage openly licensed resources such as OpenStax (MIT OpenCourseWare, 2022) and MIT OCW (Moebs et al., 2016) to generate more diverse and complex physics problems, ultimately achieving scalable and legally clear dataset generation while ensuring diverse, high-quality examples.

**Chemistry.**  Organic reaction prediction is a classic multimodal reasoning task, typically framed as symbolic input and structural output. We include a chemistry subset of 4,700 two-to-one reactions from the **USPTO-50K** dataset (Ramsundar et al., 2019), filtered for distinct reactants and single products. Each reaction trace includes three visual artifacts: individual molecular depictions of each reactant, a combined schematic of both reactants side-by-side, and the resulting product structure. Molecules are rendered with `RDKit`, and names are retrieved from PubChem when available. Text prompts use randomized templates (e.g., "What is formed by combining acetic acid and ethanol?"), and PubChem names are included when available. This visual progression helps models learn compositional chemical structure without SMILES or reaction templates.

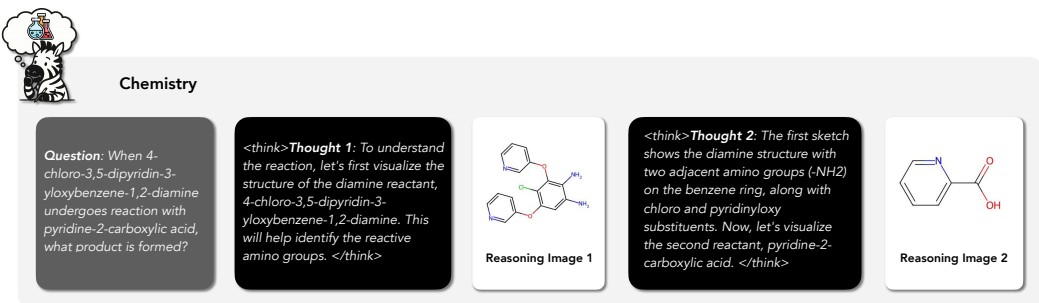

**Algorithmic problem solving.**    Humans naturally create visual diagrams when solving complex problems, transforming abstract concepts into spatial representations for deeper reasoning. We formalize this by interpreting coding problems through compact visual scaffolds: one or two diagrams depicting graph structure, edge weights, etc. To build traces, we run an iterative "visual sketchpad" loop: GPT-4.1 receives a prompt and returns `THOUGHT` statements plus `VIS_SPEC` blocks when sketches are needed; we render specs with `networkx`/`matplotlib`, feed images back to the model, and repeat until complete, then clean transcripts with post-processing.

Problem samples come from competitive programming, prioritizing real-world abstractions like logistics, network routing, and flow optimization. The orchestrator produces simple visual structures emphasizing clarity over style. Each trace contains the problem prompt, 1–3 reference diagrams, and polished explanations, supporting grounded reasoning in discrete structures while mirroring how algorithms are taught. The final corpus comprises 1,200 diverse algorithm-based problems spanning competitive programming.

**Graph problems.**    Graph algorithms are useful for large language model applications because they efficiently organize and traverse structured relationships, for example in search and retrieval applications. Methods like shortest-path and subgraph matching enable multi-step reasoning by connecting relevant concepts across knowledge graphs. Recent work by Fu et al. (2024a) shows that although LLMs can solve graph problems such as connectivity and maximum flow to some extent when a textual description of the graph is given, *multimodal* LLMs suffer when solving graph problems. This finding suggests potential for improving multimodal models' graph-understanding abilities by guiding their reasoning over images.

We create 10,000 graph problems with full reasoning traces spanning over four tasks: graph connectivity, shortest path, minimum spanning tree, and topological sort. Each task has about 2,500 samples, with one problem image and at most 19 reasoning images per sample. Each reasoning image is coupled with an explanation for the underlying algorithms, for example, Dijkstra for the shortest path, BFS for connectivity, etc.

### A.4 2D Visual Reasoning

**Visual search.** Previous research has shown that drawing bounding boxes and zooming can improve accuracy on visual search tasks (Wu and Xie, 2024; Shao et al., 2024a). We follow such tasks by creating two types of traces, one for drawing bounding boxes and one for zooming. We use data from Shao et al. (2024a) to generate our traces covering four categories of visual search tasks: chart, text/doc, relation study, and general VQA.

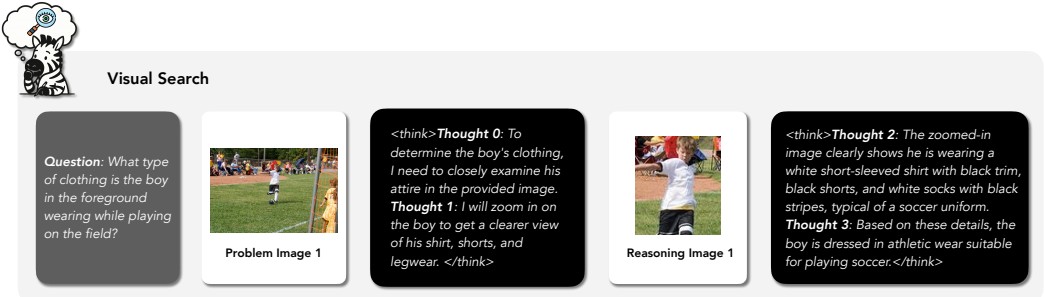

**Visual jigsaw.** Visual jigsaw refers to filling in missing pieces of an image, as in a jigsaw puzzle. Each puzzle is constructed from an ImageNet (Deng et al., 2009) image, with 1 to 4 missing pieces of varying shapes, including rectangles and irregular regions. Each puzzle includes four multiple-choice options, where each option presents a set of candidate missing pieces. Only one set correctly matches the pieces removed from the original ImageNet image. We generate two types of visual CoT traces for solving each puzzle. In the first type, we iteratively fill in the missing patches using the pieces from each multiple-choice option and identify the one that produces a coherent image. In the second type, we imagine what the original image would look like and then select the option whose pieces best match the imagined reconstruction.

### A.5 3D Visual Reasoning

**Embodied planning.** For embodied planning tasks, agents must *ground high-level decisions* in the evolving visual context of the environment. We reformulate the **ALFRED** (Shridhar et al., 2020) benchmark, an interactive 3D simulation environment where agents perform complex tasks based on human instructions, into an image goal-conditioned planning task.

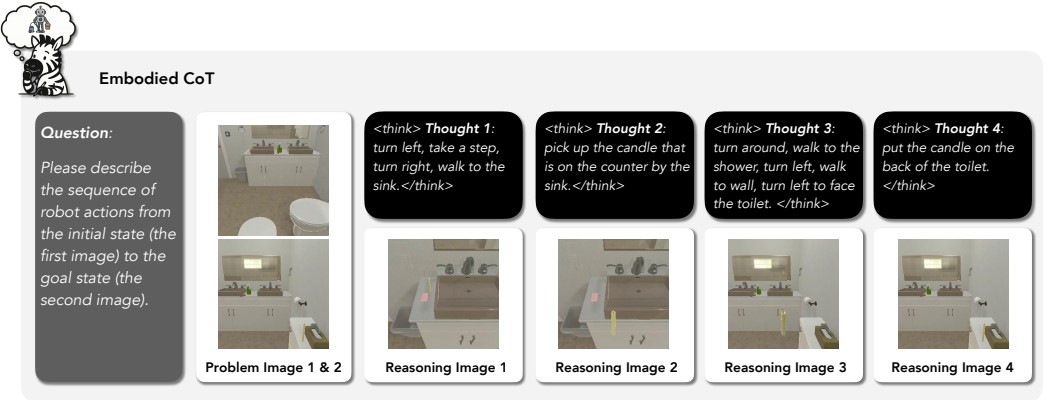

In this new task, the model receives two images: the initial and goal states. Then the model is tasked with generating a textual description of the high-level planning steps required to transition from the initial to the goal state. To emphasize the role of visual reasoning, we require the generated descriptions to be detailed and step-by-step (e.g., *"turn and go to the TV; pick up the bowl that is on the TV stand in front of the TV; with the bowl in hand..."*) rather than brief summaries (e.g., *"move bowl to coffee table"*), which can often be produced through shortcut reasoning without capturing intermediate visual steps.

We compile the entire training set, as well as the seen and unseen validation sets from ALFRED, resulting in a total of 7,080 examples spanning diverse visual reasoning trajectories. When multiple textual reasoning annotations exist for a single visual trajectory, we include all of them, resulting in 22,666 textual reasoning traces.

**Robot planning.** While low-level manipulation may rely on reactive control, continuous planning for complex tasks often requires *high-level visual guidance*, making visual CoT essential for bridging perception and long-horizon decision-making in robot planning. Similarly, we reformulate **RoboMIND** (Wu et al., 2024), a multi-embodiment dataset of real-world robot manipulation, into an image goal-conditioned planning task. In this setting, a model is provided with the initial and goal states images, along with a textual description of the robot setup (e.g., AgileX (AgileX Robotics, 2023), Franka (Franka Emika GmbH, 2018), or UR5e (Universal Robots A/S, 2018)), and is tasked with generating a detailed textual plan outlining the high-level steps required to transition from the initial to the goal state.

Unlike embodied planning tasks that often involve partial observability and require agents to infer unobserved states, this robot planning task is fully observable. Therefore, the challenge lies not in imagining the visual trajectory but in articulating precise movements for each arm or gripper to accomplish the task (e.g., *"[left] move towards the oven door and [right] grab the corn."*).

To control degrees of freedom, we exclude the humanoid robot examples from the original Robo-MIND dataset, focusing solely on tasks involving robotic arms. This results in a curated subset of 6,945 robot planning tasks, each annotated with human-generated high-level actions that serve as visual reasoning trajectories.

**3D multi-hop objects counting.** A core aspect of human visual-spatial reasoning is understanding transformations and imagining scenes from different viewpoints. For this task, our setup follows a structure similar to that of Johnson et al. (2017), using 10 predefined shape types (e.g., sphere, cylinder, donut) in various colors. At each step, we randomly apply one of three operations: remove all instances of an attribute (e.g., all red objects), remove a subset (e.g., 5 red objects), or add new objects (e.g., 2 blue prisms, 1 red sphere). We then create questions that ask about the quantity of specific attributes or what objects are left in the field. To increase difficulty, the initial scenes are rendered from varying viewpoints (front, back, left, right), where some objects may be partially occluded by those in front. The first visual reasoning step involves generating a top-down $45°$ view to reconstruct the full scene, allowing the model to see potentially blocked objects. The subsequent visual sketches correspond to each transformation step in the instruction. We also improve upon the data from Johnson et al. (2017) by adding in different materials, backgrounds, and floor designs.

## A.6 VISUAL LOGIC AND STRATEGIC GAMES

**Visual logic puzzles.** Humans approach logic puzzles such as Tetris, Raven's Progressive Matrices (RPM, Zhang et al., 2019) , and the Abstraction & Reasoning Corpus (ARC-AGI, Chollet, 2019; Chollet et al., 2024) primarily through visuospatial reasoning: we see how pieces combine, transform, or complete a pattern before committing to an answer. These logic games rely heavily on visuospatial working memory, which is correlated with general intelligence level (Lau-Zhu et al., 2017; de Winter et al., 2023).

To enhance models with such cognitive ability, we include the following tasks. For *Tetris*, we collect three types of tasks: a) shape assembly: given a silhouette and candidate tetromino sets, select the one that perfectly tiles the shape; b) grid completion: fill a partially occupied grid using a specified set of tetrominoes; c) spatial transformation: apply a sequence of geometric operations (translate, rotate, mirror, scale) to an irregular shape in the grids. The visual CoT involves visualizing each transformation step. For *RPM* (IQ matrix), we include three types from Zhang et al. (2019) that

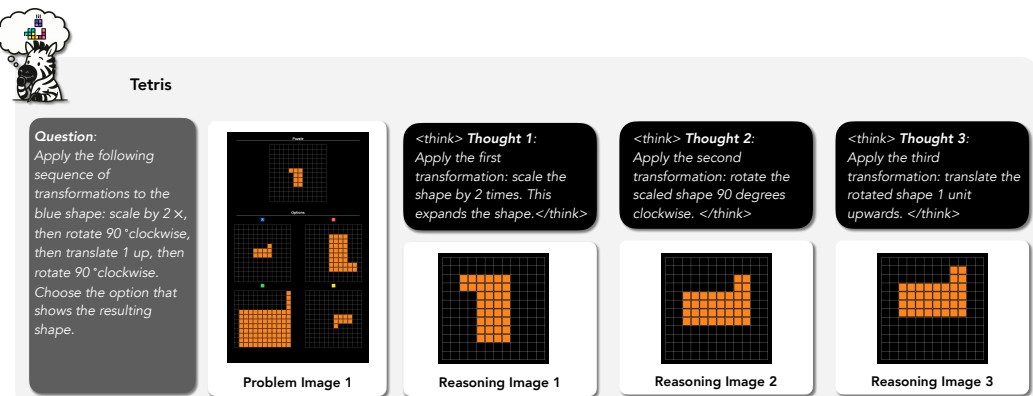

involve compositional reasoning. The reasoning trace identifies visual patterns for each compositional component across rows or columns. For *ARC-AGI*, while prior models often rely on textual reasoning, humans typically solve these tasks through visual pattern recognition. To better align with human strategies, we construct two types of visual CoT. The first begins with matrix representations of the training examples and test input; the reasoning trace first visualizes the training examples, the test input, and finally the predicted output. The second type directly uses visual representations in the task instruction, thus the model only has to generate a visual sketch of the predicted output as part of its reasoning process. For all data, we use VLM to generate accompanying textual descriptions to enrich interleaved text-image rationales.

**Mazes.** Mazes serve as a canonical testbed for visual CoT reasoning, bridging low-level perception with high-level symbolic search. Unlike purely pixel-based 2D visual tasks such as visual search and visual jigsaw, mazes possess explicit graph structure yet remain visually intuitive, letting us disentangle vision errors from planning errors.

We adopt the `maze-dataset` library to procedurally generate thousands of grid mazes with diverse topologies (lattice type, branch factor, loop density).[1] Each instance is exported in two complementary formats: a) `m.as_pixels()`, an RGB raster that encodes walls, free cells, start ■, and goal ■, suitable for visual perception; b) `MazePlot`, a vector overlay that can superimpose solution paths, candidate trajectories, heat-maps, or landmark nodes for human-readable walk-throughs. To increase maze diversity, we also use OpenAI Gym's `FrozenLake-v1` environment (Brockman et al., 2016) .

We evaluate a broad spectrum of spatial reasoning skills across multiple question types: *I. topological analysis* (e.g., counting isolated regions, identifying connected components under 4- or 8-connectivity, finding the largest connected area), *II. pathfinding* (e.g., determining reachable endpoints, computing shortest paths, enumerating all optimal routes), *III. navigation planning* (e.g., selecting correct paths from alternatives, calculating minimal moves to reach targets), and *IV. coverage problems* (e.g., visiting all marked locations, identifying the farthest reachable position). This diverse task suite goes beyond simple start-to-goal navigation, encompassing the full range of spatial reasoning strategies that humans use to interpret complex environments. We also introduce varying complexity of the matrix, including different maze side lengths ranging from $(5, 15)$, different branching factors $b$, loop probability $\ell$, and number of distractor endpoints $k$. Larger $n$ exponentially increases the search space, while higher $b$ and $\ell$ degrade heuristic admissibility. Both of those require genuine planning rather than rote memorization.

**Chess.**

Strategic planning in chess involves simulating multiple futures and selecting moves that maximize long-term advantage. To support counterfactual reasoning, we construct a dataset of mid-game positions from rated Lichess games [2], each with structured visual traces. Given a position, `Stockfish`

---

[1] `maze-dataset` supports recursive-backtracker, randomized Prim, Wilson, and Kruskal generators; see (Ivanitskiy et al., 2023).

[2] https://lichess.org/

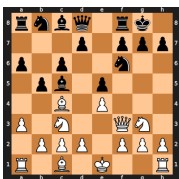

What's White's best move? Options: A: Ba2, B: Na4, C: Qf5, D: Bb3.

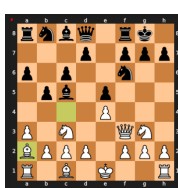 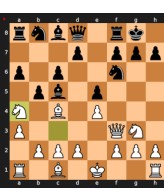 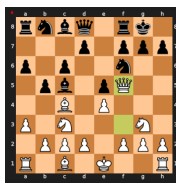 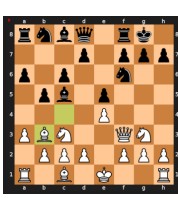

(A) **Ba2**: Safe position supports central pawns    (B) **Na4**: Poorly placed weak attack on b6    (C) **Qf5**: Exposed queen vulnerable to g6    (D) **Bb3**: Vulnerable position weaker than Ba2

Figure 7: Traces showing reasoning for each move option. Option A (Ba2) is evaluated as strongest, providing safe bishop placement while supporting potential central pawn advances.

identifies the optimal move, and three alternates are sampled randomly from legal moves. Each candidate is visualized independently for comparative evaluation. By rendering possibilities in isolation, move consequences, tempo gain, structural weakening, and tactical motifs become legible, enabling better strategic reasoning. Traces are formatted as multiple-choice tasks with visual sketches, encouraging tactical awareness and pattern recognition. Postprocessing with Gemini 2.5 Flash refines traces into coherent multimodal reasoning sequences for short and long-horizon planning, yielding 11,500 chess traces across diverse positions and hypothetical move states.

**Connect 4.** We also include 2,100 Connect-Four traces that teach short-horizon grid planning. Each starts from a mid-game position obtained after 4–20 random moves; a depth-4 $\alpha$–$\beta$ minimax (center-control, open-three heuristic) then plays the forced win. Every drop is rendered on a $7 \times 6$ board with a glowing overlay, and an LLM condenses the raw log into a clear, step-by-step "How does Red win?" narrative pairing text and visuals.

**Checkers.** Our checkers traces, initiated from randomized mid-game boards and continued through depth-4 minimax, are designed to capture key tactical motifs such as forced captures, multi-jump chains, and king-row advancement, providing dense supervision for spatially grounded decision making. To ensure data quality, we apply rule-based validators that discard invalid traces. Final traces are passed through a language model to synthesize coherent visual reasoning aligned with *turn-based planning*, ending with 2,800 full traces.

**Ciphers.** Classical encryption schemes convert symbolic rules into spatial transformations, enabling visual reasoning over grids and coordinates. We generate examples of *Scytale*, *Polybius*, *Rail Fence* and *Playfair* ciphers, each with multimodal traces showing encoding steps. Visual sketches highlight relevant regions such as column positions, grid lookups, or digraph geometry, while text describes applied rules (e.g., *"Letter 'P' goes on the second rail," "'A' is at coordinates (1, 1)."*) Early steps use visuals, later steps use language, mirroring real cryptanalysis. The 6,600 traces align symbolic manipulation with visual structure.

## B    EXTRA BAGEL ZEBRA COT GENERATED VISUAL COT

---

### Visual CoT Example: EMMA

**Prompt:**

There are 81 grid points (uniformly spaced) in the square shown in the diagram below, including the points on the edges. Point $P$ is the center of the square. Given that point $Q$ is randomly chosen from among the other 80 points, what is the probability that line $PQ$ is a line of symmetry for the square?

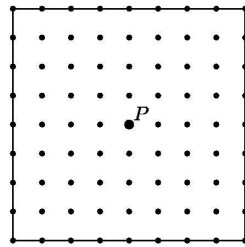

---

**Visual CoT:**

**\<think\>**

The problem asks for the probability that a randomly chosen point $Q$ from the 80 points other than the center $P$ results in the line $PQ$ being a line of symmetry for the square. We need to determine the number of such points $Q$ and divide by the total number of possible points (80).

First, let's visualize the grid and the center point $P$. The grid has 9 points along each side, so there are 80 points in total (excluding $P$). A line of symmetry for a square passes through the center $P$ and divides the square into two congruent halves. For $PQ$ to be a line of symmetry, $Q$ must be the reflection of $P$ across this line. This means $Q$ must be on the grid line that forms a line of symmetry with $P$.

Let's consider the lines of symmetry for the square. A square has 4 lines of symmetry: 2 diagonals, the vertical line through the center, and the horizontal line through the center. Let's visualize these lines.

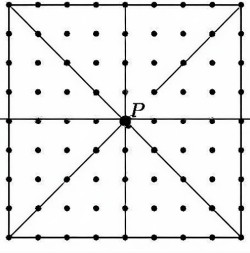

**[Visual Thought]**

**\</think\>**

---

## Visual CoT Example: VisuLogic

**Prompt:**

Solve the complex visual logical reasoning problem through step-by-step reasoning.

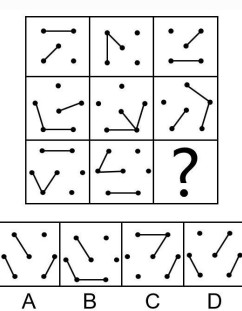

**Visual CoT:**

**<think>**
Observe the grid to identify patterns in the shapes and their components. The grid is divided into 3x3 cells. Each cell contains two distinct shapes: a line segment and a dot. I will analyze the patterns for the line segment and the dot separately. For the line segment, the pattern is: (Top Left, Top Right) = (Bottom Left, Bottom Right). For the dot, the pattern is: (Top Left, Top Right) = (Bottom Left, Bottom Right). Now, I will apply these patterns to the third row and third column to find the missing shape. The missing shape in the third row and third column should have a line segment like the one in the third row, third column (which is the same as the third row, second column line segment). The missing

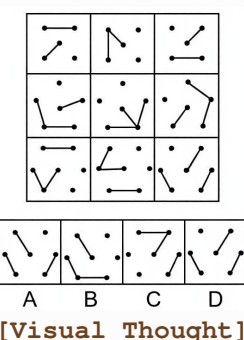

**[Visual Thought]**

shape in the third row and third column should also have a dot like the one in the third row, third column (which is the same as the third row, second column dot). Combining these, the missing shape should be a line segment from the third row, second column and a dot from the third row, second column.
**</think>**

## Visual CoT Example: EMMA

### Prompt:

The picture shows a cube with four marked angles: $\angle WXY$, $\angle XYZ$, $\angle YZW$, and $\angle ZWX$. What is the sum of these angles?

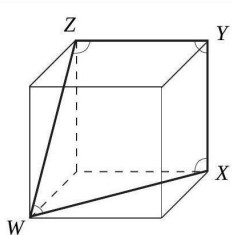

### Visual CoT:

<think>
The problem asks for the sum of four specific angles in a cube. The angles are $\angle WXY$, $\angle XYZ$, $\angle YZW$, and $\angle ZWX$. These angles are formed by the edges of the cube meeting at vertices $X, Y, Z, W$.

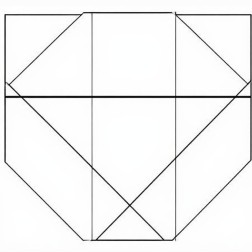

In a cube, each vertex is formed by three edges that are mutually perpendicular. This means that at each vertex, the three edges meet to form right angles.

Let's consider vertex $X$. The edges $WX$, $XY$, and $XZ$ meet at vertex $X$. Since the edges are perpendicular, the angle $\angle WXY = 90°$.

Similarly, we can analyze the other vertices:

**[Visual Thought]**

- At vertex $Y$: edges $XY$, $YZ$ are perpendicular, so $\angle XYZ = 90°$

- At vertex $Z$: edges $YZ$, $ZW$ are perpendicular, so $\angle YZW = 90°$

- At vertex $W$: edges $ZW$, $WX$ are perpendicular, so $\angle ZWX = 90°$

We have identified the measures of all four angles: $\angle WXY = 90°$, $\angle XYZ = 90°$, $\angle YZW = 90°$, $\angle ZWX = 90°$

To find the sum of these angles, we add their measures:

$$90° + 90° + 90° + 90° = 360°$$

The sum of the angles $\angle WXY$, $\angle XYZ$, $\angle YZW$, and $\angle ZWX$ is **360°**.
</think>

**Visual CoT Example: MathVista**

**Prompt:**

Question: As shown in the figure, $AB$ is the diameter of $\odot O$, and point $C$ is on $\odot O$. If $\angle A = 40.0$, then the degree of $\angle B$ is?

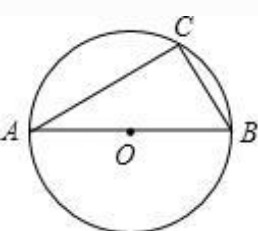

**Visual CoT:**

**<think>**
First, let's analyze the geometry of the problem. Since AB is the diameter of the circle, any angle inscribed in the semicircle formed by AB will be a right angle. Thus, angle ACB is a right angle, measuring 90 degrees. Now, let's draw an auxiliary line connecting point O (the center of the circle) to point C. This line segment OC is a crucial part of our reasoning. The line segment OC can be labeled as shown in the image.

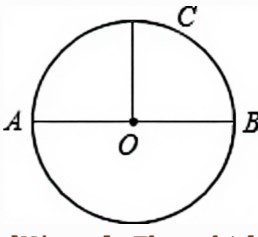

**[Visual Thought]**

Since AB is the diameter of the circle, O is the center. Therefore, AO and BO are radii of the circle, making them equal in length. By the properties of an isosceles triangle (AO = BO), and having an angle of 90 degrees at C (i.e., angle ACB), we can deduce that triangle ABC is an isosceles right-angled triangle. This means that angle A is equal to angle CBA, and since angle ACB is 90 degrees, the two other angles must add up to 90 degrees. Given that angle A is 40 degrees, angle B must be 50 degrees. Therefore, angle B is 50 degrees.
**</think>**

## Visual CoT Example: BLINK

**Prompt:**

How many kids are sitting on the bench in the image?

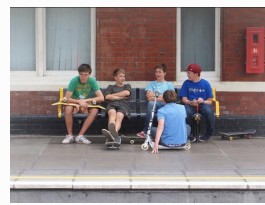

---

**Visual CoT:**

**<think>**

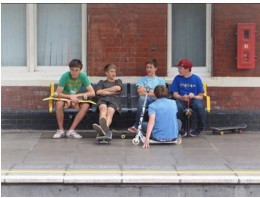

**[Visual Thought]**

Let's count the number of kids sitting on the bench in the image. To get a clearer view of the kids on the bench, I will zoom in on the relevant area. The zoomed-in image clearly shows four children sitting on the bench. Based on this visual evidence, the number of kids on the bench is four. **</think>**

# C  ANOLE-ZEBRA PERFORMANCE BREAKDOWN

| Split | Anole | Anole-Zebra-CoT (Ours) |
|---|---|---|
| Overall | 12.80 | **15.03** |
| Chemistry | 12.84 | **15.48** |
| Coding | 9.75 | **16.31** |
| Math | 13.12 | **14.35** |
| Physics | **21.79** | 10.90 |

Table 4: EMMA: breakdown by subject (%).

| Subtask | Anole | Anole-Zebra-CoT (Ours) |
|---|---|---|
| Overall | 22.80 | **24.90** |
| Scientific reasoning | 30.33 | **32.79** |
| Textbook question answering | **36.08** | 29.75 |
| Numeric commonsense | 16.67 | **17.36** |
| Arithmetic reasoning | 15.58 | **18.98** |
| Visual question answering | 24.58 | **29.61** |
| Geometry reasoning | 20.50 | **23.01** |
| Algebraic reasoning | **25.27** | 24.56 |
| Geometry problem solving | 21.15 | **24.04** |
| Math word problem | 9.14 | **12.37** |
| Logical reasoning | **29.73** | 10.81 |
| Figure question answering | 24.54 | **28.25** |
| Statistical reasoning | 20.27 | **26.58** |

Table 5: MathVista: breakdown by subtask for base vs. our model (%).

| Subtask | Anole | Anole-Zebra-CoT (Ours) |
|---|---|---|
| Overall | 8.50 | **21.80** |
| Quantitative reasoning | 8.78 | **21.81** |
| Spatial reasoning | 8.23 | **22.08** |
| Positional reasoning | 8.82 | **19.85** |
| Attribute reasoning | 9.76 | **25.61** |
| Stylistic reasoning | 10.00 | **24.44** |
| Other | 5.56 | **18.52** |

Table 6: Visual Logic: breakdown by subtask (%).

| Category | Anole | Anole-Zebra-CoT (Ours) |
|---|---|---|
| Overall | 26.46 | **31.25** |
| Art Style | 19.66 | **35.04** |
| Counting | **19.17** | 15.00 |
| Forensic detection | 0.00 | **20.45** |
| Functional correspondence | 17.69 | **22.31** |
| IQ test | **26.67** | 23.33 |
| Jigsaw | 11.33 | **39.33** |
| Multi-view reasoning | **48.12** | 21.05 |
| Object localization | **50.82** | 45.90 |
| Relative depth | 38.71 | **41.94** |
| Relative reflectance | **29.10** | 27.61 |
| Semantic correspondence | **19.42** | 17.99 |
| Spatial relation | 41.26 | **57.34** |
| Visual correspondence | 21.51 | **26.16** |
| Visual similarity | 30.37 | **44.44** |

Table 7: Blink: breakdown by category (%).

# D    BAGEL PERFORMANCE ANALYSIS

We evaluate our Bagel model trained on ZEBRA-COT across several benchmarks but did not observe substantial improvements over the original model, where the original generates pure text responses. In fact, we even saw slight performance drops on some tasks such as MathVista. A detailed analysis revealed a likely cause of this decline. The Bagel model employs two visual encoders: a ViT-based understanding encoder and a VAE-based generation encoder. For generated images, the model often produces hallucinations. For example, when instructed to remove all red balls from a scene, the generated image may also remove yellow balls. When this corrupted image is passed back into the ViT encoder, the representation correctly reflects that both red and yellow balls are missing, leading the model to reason over inaccurate visual information, ultimately reducing accuracy. Instead generating pure text responses avoids such image generation hallucinations.

# E  SCAFFOLDING RESULTS BREAKDOWN

| Chess | | | | | |
| --- | --- | --- | --- | --- | --- |
| Model | Q (%) | 1MT (%) | 2MT (%) | Δ1MT (%) | Δ2MT (%) |
| Claude-4 Sonnet | 32.95 | 57.95 | 67.05 | 25.00 | 34.09 |
| Gemini-2.5 Pro | 15.07 | 39.73 | 39.73 | 24.66 | 24.66 |
| GPT-5 | 45.78 | 62.65 | 61.45 | 16.87 | 15.66 |

| Graph | | | | | |
| --- | --- | --- | --- | --- | --- |
| Model | Q (%) | 1MT (%) | 2MT (%) | Δ1MT (%) | Δ2MT (%) |
| Claude-4 Sonnet | 8.11 | 20.72 | 22.52 | 12.61 | 14.41 |
| Gemini-2.5 Pro | 1.90 | 11.43 | 20.95 | 9.52 | 19.05 |
| GPT-5 | 1.74 | 14.78 | 22.61 | 13.04 | 20.87 |

| 2D Visual Jigsaw | | | | | |
| --- | --- | --- | --- | --- | --- |
| Model | Q (%) | 1MT (%) | 2MT (%) | Δ1MT (%) | Δ2MT (%) |
| Claude-4 Sonnet | 21.74 | 36.23 | 62.32 | 14.49 | 40.58 |
| Gemini-2.5 Pro | 34.38 | 56.25 | 59.38 | 21.88 | 25.00 |
| GPT-5 | 62.86 | 77.14 | 85.71 | 14.29 | 22.86 |

| Maze | | | | | |
| --- | --- | --- | --- | --- | --- |
| Model | Q (%) | 1MT (%) | 2MT (%) | Δ1MT (%) | Δ2MT (%) |
| Claude-4 Sonnet | 35.06 | 58.44 | 94.81 | 23.38 | 59.74 |
| Gemini-2.5 Pro | 59.70 | 85.07 | 97.01 | 25.37 | 37.31 |
| GPT-5 | 63.01 | 86.30 | 97.26 | 23.29 | 34.25 |

| 3D Multi-Hop Counting | | | | | |
| --- | --- | --- | --- | --- | --- |
| Model | Q (%) | 1MT (%) | 2MT (%) | Δ1MT (%) | Δ2MT (%) |
| Claude-4 Sonnet | 59.68 | 67.74 | 74.19 | 8.06 | 14.52 |
| Gemini-2.5 Pro | 45.95 | 48.65 | 51.35 | 2.70 | 5.41 |
| GPT-5 | 72.58 | 74.19 | 77.42 | 1.61 | 4.84 |

| Tetris | | | | | |
| --- | --- | --- | --- | --- | --- |
| Model | Q (%) | 1MT (%) | 2MT (%) | Δ1MT (%) | Δ2MT (%) |
| Claude-4 Sonnet | 18.87 | 39.62 | 45.28 | 20.75 | 26.42 |
| Gemini-2.5 Pro | 8.57 | 25.71 | 31.43 | 17.14 | 22.86 |
| GPT-5 | 30.77 | 50.00 | 59.62 | 19.23 | 28.85 |

Table 8: Scaffolding evaluation results across task domains. Q: zero-shot question-only; 1MT: question with first multimodal reasoning step; 2MT: question with first two multimodal reasoning steps. Δ columns show absolute improvement over baseline (Q).

| Chess | | | | |
|---|---|---|---|---|
| Model | Q (%) | 1TT (%) | 2TT (%) | Δ1TT (%) | Δ2TT (%) |
| Claude-4 Sonnet | 32.95 | 42.05 | 40.91 | 9.10 | 7.96 |
| Gemini-2.5 Pro | 15.07 | 13.70 | 19.86 | -1.37 | 4.79 |
| GPT-5 | 45.78 | 51.81 | 62.65 | 6.03 | 16.87 |

| Graph | | | | |
|---|---|---|---|---|
| Model | Q (%) | 1TT (%) | 2TT (%) | Δ1TT (%) | Δ2TT (%) |
| Claude-4 Sonnet | 8.11 | 20.72 | 15.32 | 12.61 | 7.21 |
| Gemini-2.5 Pro | 1.90 | 5.71 | 5.71 | 3.81 | 3.81 |
| GPT-5 | 1.74 | 11.30 | 19.13 | 9.56 | 17.39 |

| 2D Visual Jigsaw | | | | |
|---|---|---|---|---|
| Model | Q (%) | 1TT (%) | 2TT (%) | Δ1TT (%) | Δ2TT (%) |
| Claude-4 Sonnet | 21.74 | 39.13 | 30.43 | 17.39 | 8.69 |
| Gemini-2.5 Pro | 34.38 | 24.22 | 25.00 | -10.16 | -9.38 |
| GPT-5 | 62.86 | 68.57 | 71.43 | 5.71 | 8.57 |

| Maze | | | | |
|---|---|---|---|---|
| Model | Q (%) | 1TT (%) | 2TT (%) | Δ1TT (%) | Δ2TT (%) |
| Claude-4 Sonnet | 35.06 | 49.35 | 55.84 | 14.29 | 20.78 |
| Gemini-2.5 Pro | 59.70 | 23.29 | 39.73 | -36.41 | -19.97 |
| GPT-5 | 63.01 | 63.01 | 75.34 | 0.00 | 12.33 |

| 3D Multi-Hop Counting | | | | |
|---|---|---|---|---|
| Model | Q (%) | 1TT (%) | 2TT (%) | Δ1TT (%) | Δ2TT (%) |
| Claude-4 Sonnet | 59.68 | 69.35 | 69.35 | 9.67 | 9.67 |
| Gemini-2.5 Pro | 45.95 | 54.84 | 56.45 | 8.89 | 10.50 |
| GPT-5 | 72.58 | 72.58 | 74.19 | 0.00 | 1.61 |

| Tetris | | | | |
|---|---|---|---|---|
| Model | Q (%) | 1TT (%) | 2TT (%) | Δ1TT (%) | Δ2TT (%) |
| Claude-4 Sonnet | 18.87 | 26.42 | 32.08 | 7.55 | 13.21 |
| Gemini-2.5 Pro | 8.57 | 11.54 | 7.69 | 2.97 | -0.88 |
| GPT-5 | 30.77 | 28.85 | 42.31 | -1.92 | 11.54 |

Table 9: Text-only CoT evaluation results across task domains. Q: zero-shot question; 1TT: first text reasoning step; 2TT: first two text reasoning steps. Δ columns show absolute improvement over Q.

# F PROMPT TEMPLATES

## F.1 PROMPT FOR ENHANCING RAW REASONING TRACES FOR ONLINE AND AGENTIC DATA

---

**Prompt Template 1**

```
You are an expert in creating clean and logically coherent
↪   multimodal chain of thought traces. Your task is to
↪   analyze
and comprehend a raw reasoning trace with interleaved text
↪   and images, then transform it into a clean, step-by-step
↪   multimodal
reasoning trace that correctly solves the original problem.

====================== INPUT ========================
1. Problem & Noisy Trace: A raw interleaved text and image
↪   reasoning trace. Images in this trace are represented by
↪   placeholders:
   - `[problem_image_X]` for original problem images (e.g.,
     ↪   `[problem_image_1]`, `[problem_image_2]`)
   - `[reasoning_image_X]` for images generated during
     ↪   reasoning (e.g., `[reasoning_image_1]`,
     ↪   `[reasoning_image_2]`)
2. Image Data: The actual image data corresponding to the
↪   placeholders, provided separately.

===================== Your Task ==================
Generate a clean, logical multimodal reasoning trace as
↪   **plain text** that represents the *ideal* reasoning
↪   process to solve the problem.

==================== OUTPUT FORMAT ===================
You MUST generate the formatted reasoning trace with the
↪   following structure:

QUESTION:
<The original problem statement with text and image
↪   placeholders: <image_start>[problem_image_1]<image_end>,
↪   <image_start>[problem_image_2]<image_end>, etc. Stay as
↪   close to the original problem statement as possible but
↪   remove noise to ensure clarity>

REASONING TRACE:
THOUGHT 0: <Clear description of initial reasoning step that
↪   identifies key elements of the problem>
THOUGHT 1: <Next reasoning step, often explaining why an
↪   image will be created>
<image_start>[reasoning_image_1]<image_end>
THOUGHT 2: <Further reasoning step based on the image,
↪   explaining insights gained>
<image_start>[reasoning_image_2]<image_end>
// Additional thoughts and images as needed
<image_start>[reasoning_image_X]<image_end>
THOUGHT N: <Final reasoning step before the answer,
↪   summarizing key insights>

FINAL ANSWER:
```

```
<The final calculated answer based on the reasoning>

===================== Guidelines  =================

1. Enhancing Original Trace Rather than Generating New Trace:
   - Instead of generating a new trace, your task is to
     ↪  enhance the original trace (which is a correct trace
     ↪  but rather concise and lacks coherent multimodal
     ↪  reasoning) by adding more details and explanations, see
     ↪  the following sections of guidelines for more details.
   - You MUST use all the images provided in the original
     ↪  trace.
   - You should use the original trace as a reference rather
     ↪  than copying it verbatim.

2. Multimodal Reasoning Flow:
   - Develop a coherent, step-by-step chain of thought that
     ↪  seamlessly integrates textual and visual reasoning.
   - Clearly explain the necessity of generating a sketch /
     ↪  visual thought / image before introducing its
     ↪  placeholder.
   - After each image placeholder, describe the insights
     ↪  gained from the sketch / visual thought / image, and
     ↪  how it contributes to advancing the solution.
   - Ensure each step logically builds on the previous ones,
     ↪  especially between text reasoning and visual reasoning
   steps.

3. Image Placeholders and References:
   - Use placeholder tags ONLY when you want to actually
     ↪  insert/show/generate an image in your trace. When
     ↪  doing so, write the corresponding placeholder tag
     ↪  exactly as shown, including the <image_start> and
     ↪  <image_end> tags.
   - Each unique image in the original problem and the
     ↪  reasoning trace should be represented by a unique
     ↪  placeholder tag, and each unique placeholder tag
     ↪  should only show up once in the trace.
   - When referring to images in your explanations, use
     ↪  natural language descriptions (e.g., "the diagram in
     ↪  the question", "the first sketch", "the visual thought
     ↪  X I created") instead of using placeholder tags. This
     ↪  is important because it helps us to parse into
     ↪  interleaved text and image sequences.
   - For images from the original problem, use:
     ↪  <image_start>[problem_image_X]<image_end>
   - For sketches or visuals generated during reasoning, use:
     ↪  <image_start>[reasoning_image_X]<image_end>

4. Narrative Style:
   - Remove irrelevant technical details such as debugging
     ↪  info, code snippets, and LaTeX package imports.
   - Eliminate verbose language that do not contribute to
     ↪  solving the problem.
   - Focus on the essential reasoning path that leads to the
     ↪  correct solution, using concise and clear language to
     ↪  describe the overall reasoning process.
```

## F.2 Prompt for Enhancing Program Generated Template Data

---

**Prompt Template 1**

```
You are an expert in enhancing multimodal reasoning traces.
↪  Your task is to transform a template reasoning trace into
↪  a diverse multimodal reasoning trace that correctly
↪  solves the problem, while staying close to the original
↪  template and final answer.

======================  INPUT  ======================
1. Problem & Template Trace: A template with interleaved text
↪  and image placeholders:
    - `[problem_image_X]` for original problem images (e.g.,
    ↪  `[problem_image_1]`)
    - `[reasoning_image_X]` for images generated during
    ↪  reasoning (e.g., `[reasoning_image_1]`)
2. Image Data: The actual image data corresponding to the
↪  placeholders, provided separately.

====================  Your Task  =================
Generate a concise multimodal reasoning trace as **plain
↪  text**.

====================  OUTPUT FORMAT  ==================
You MUST generate the formatted reasoning trace with the
↪  following structure:

QUESTION:
<Rewrite the problem statement in your own words while
↪  maintaining all key information. Do not change key
↪  information. Include image placeholders:
↪  <image_start>[problem_image_1]<image_end>,
↪  <image_start>[problem_image_2]<image_end>, etc.>

REASONING TRACE:
THOUGHT 0: <Identify key elements of the problem>
THOUGHT 1: <Explain reasoning step, often why an image /
↪  sketch / visual thought is needed>
<image_start>[reasoning_image_1]<image_end>
THOUGHT 2: <Explain insights from the image>
<image_start>[reasoning_image_2]<image_end>
// Additional thoughts and images as needed
<image_start>[reasoning_image_X]<image_end>
THOUGHT N: <Summarize key insights before answer>

FINAL ANSWER:
<The original final answer in the template, do not change it>

====================  Guidelines  ==================

1. Diversifying the Template:
    - Rewrite the problem statement and reasoning steps in
    ↪  your own words while preserving all key information.
    - Avoid deviating from the original template reasoning
    ↪  structure. Your job is to diversify the text of the
    ↪  original trace, not the logic.
```

```
      - Vary the language and phrasing to avoid repetitive
      ↪  patterns.
      - You MUST use all the images provided in the original
      ↪  trace.
      - You MUST keep the original final answer.
      - Maintain the original template's core reasoning
      ↪  structure and rationale while introducing textual
      ↪  reasoning refinements rather than substantial changes
      ↪  to the logical flow.

2. Multimodal Reasoning Flow:
      - Develop a coherent, step-by-step chain of thought that
      ↪  seamlessly integrates textual and visual reasoning.
      - Clearly explain the necessity of generating a sketch /
      ↪  visual thought / image before introducing its
      ↪  placeholder.
      - After each image placeholder, describe the insights
      ↪  gained from the sketch / visual thought / image, and
      ↪  how it contributes to advancing the solution.
      - Ensure each step logically builds on the previous ones,
      ↪  especially between text reasoning and visual reasoning
      ↪  steps.

3. Image Placeholders and References:
      - Use placeholder tags ONLY when you want to actually
      ↪  insert/show/generate an image in your trace. When
      ↪  doing so, write the corresponding placeholder tag
      ↪  exactly as shown, including the <image_start> and
      ↪  <image_end> tags.
      - Each unique image in the original problem and the
      ↪  reasoning trace should be represented by a unique
      ↪  placeholder tag, and each unique placeholder tag
      ↪  should only show up once in the trace.
      - When referring to images in your explanations, use
      ↪  natural language descriptions (e.g., "the diagram in
      ↪  the question", "the first sketch", "the visual thought
      ↪  X I created") instead of using placeholder tags. This
      ↪  is important because it helps us to parse into
      ↪  interleaved text and image sequences.
      - For images from the original problem, use:
      ↪  <image_start>[problem_image_X]<image_end>
      - For sketches or visuals generated during reasoning, use:
      ↪  <image_start>[reasoning_image_X]<image_end>
```

## F.3 PROMPT FOR ALGORITHMIC PROBLEMS

**Prompt Template 2**

```
You are an expert in mathematical problem solving,
  ↪  algorithmic reasoning, visual explanation, and creating
  ↪  multimodal reasoning traces.

---
1. STRICT VISUALIZATION POLICY (IMPORTANT):
```

```
You are only allowed to produce at most 3 [VIS_SPEC]
↪  visualizations, and they must all appear at the very
↪  beginning of your reasoning (within the first 3--4
↪  thoughts). You may only use the following types for these
↪  visualizations:
   – graph
   – flow_network
   – tree_from_dict
   – tree_from_root
   – grid

After these initial visualizations, you must do all further
↪  reasoning purely mentally/textually or with
↪  pseudocode--NO MORE [VIS_SPEC] blocks are allowed after
↪  the first 3. Any attempt to include more than 3
↪  visualizations or use a disallowed type will be ignored.
The visual reasoning should only be used to understand the
↪  setup of the question – humans visualize at the beginning
↪  to ``set the board.'' The actual problem solving is done
↪  purely textually.

**General Rules:**
- Interleave THOUGHT steps and [VIS_SPEC] image requests.
- Your final reasoned solution must match the logic of the
↪  given solution code.
- Prefix THOUGHT 0 with REASONING TRACE in the previous line.
- Prefix each reasoning step with ``THOUGHT n:'' (n starts at
↪  0, less than 50 words each).
- Max 3 [VIS_SPEC] blocks, all within the first 3--4
↪  thoughts.
   – Diagram #1: raw structural sketch (graph topology, blank
    ↪  grid, etc.).
   – Diagram #2--3: showcase pivotal elements if helpful.
- **Internal self-check (no output):** ``Would a human
↪  scribble this as a quick setup sketch?'' If the answer is
↪  no, **do not** emit a VIS_SPEC.
- Strictly do not regenerate the same image – simply refer to
↪  the previous images in text if needed.
- Max of 10 thoughts.
- Every visualization request **must** use a minimal
↪  [VIS_SPEC] block with the correct type specified. Do not
↪  use any other format.
- Do **not** include file names, imports, or drawing code.
↪  The orchestrator will handle image generation.
- If you cannot meaningfully visualize or correctly visualize
↪  a thought using the provided tools and inputs, then do
↪  not generate an image.
- Images are meant to be simple and visually cohesive – do
↪  not make grandiose images with titles and axis – it's
↪  simply for a baseline understanding of the question.
- The first line of the trace should be QUESTION: followed by
↪  a detailed in depth recap of the problem, specifying all
↪  the important aspects, without mentioning the solution.

2. Validity Rules:
```

```
- All [VIS_SPEC] parameters must be valid, fully-formed
  ↪ Python literals.
- For [VIS_SPEC] type "grid", the values must be a valid
  ↪ Python list of lists with exactly rows rows and cols
  ↪ columns (or a flat list of length rows * cols), and each
  ↪ value should be a number or string.
- For type graph, tree_from_dict, tree_from_root, and
  ↪ similar, node and edge labels may be strings or integers,
  ↪ but all structures must be valid Python literals.
- Never output incomplete or empty lists/arrays/dicts in
  ↪ [VIS_SPEC] blocks. All lists must be fully closed and
  ↪ contain at least one value, unless an empty structure is
  ↪ explicitly required by the problem.
- Do not use variable names, symbolic labels, ellipses, or
  ↪ placeholders (e.g., a1, x, \ldots, an) anywhere in the
  ↪ [VIS_SPEC].

---

**[VIS_SPEC] Reference Examples: Your blocks must follow the
  ↪ same format as these.**

[VIS_SPEC]
type: graph
nodes: [A,B,C]
edges: [(A,B),(B,C)]
[/VIS_SPEC]

[VIS_SPEC]
type: flow_network
nodes: [A,B,C]
edges: [(A,B),(B,C)]
flows (optional): {(A,B): 2, (B,C): 1}
capacities (optional): {(A,B): 3, (B,C): 2}
[/VIS_SPEC]

\ldots
\ldots
\ldots

3. Reflection step immediately after each VIS_SPEC
   - Write a new THOUGHT that:
     a. Describes what you see in the previous generated
        ↪ `reasoning_image_N.png`.
     b. Explains how it informs your next reasoning move.

4. FINAL ANSWER
   - After all reasoning, output ``FINAL ANSWER:'' and your
     ↪ concise solution (pseudocode is sufficient)

5. Formatting and Output Requirements
   - Everything must be plain text with only the full
     ↪ QUESTION (just the problem itself, not the name of the
     ↪ problem), FINAL ANSWER, REASONING TRACE marker,
     ↪ THOUGHT lines and VIS_SPEC markers.
```

## G    IMPACT STATEMENT

All data sourced in this work were either publicly available under open licenses or generated synthetically. We ensured that all original content and assets used in the dataset creation process respect copyright and licensing terms. No human subjects were involved, and we do not foresee any direct harm to individuals or communities as a result of this work. The dataset is intended solely for academic research to improve multimodal reasoning capabilities in AI systems.

## H    LICENSES

We list the licenses involved in this work as follows,

- `Anole-7B` model is under *Chameleon Research License*.
- `BAGEL-7B-MoT` model is licensed under the *Apache 2.0 license*. It is finetuned from *Qwen2.5-7B-Instruct* and *siglip-so400m-14-384-flash-attn2* model, and uses the *FLUX.1-schnell VAE model*, all under *Apache 2.0*.
- `ImageNet` dataset in under *BSD 3* license.
- `Visual CoT` dataset is licensed under *CC BY 4.0*
- `MATH` dataset (Hendrycks et al., 2021) is under *MIT License*.
- `OpenStax` Physics books are license under *CC BY 4.0*.
- `MIT OCW` Physics lecture notes under *CC BY 4.0*.
- `Maze` datasets is licensed under *CC BY 4.0*.

