# OpenReview forum: "Zebra-CoT: A Dataset for Interleaved Vision-Language Reasoning"
_ICLR.cc/2026/Conference — ICLR 2026 Poster_

### Official Review · Reviewer_UEnu · 2025-10-23

**Soundness:** 4
**Presentation:** 3
**Contribution:** 3
**Rating:** 6
**Confidence:** 3

**Summary:**

The authors propose a visual chain-of-thought (CoT) collection methods that can provide visual CoT trajectories with strong correlation between the text and the images. They use the methods to collect 182k trajectories across 18 domains. They authors also fine-tune two models to support its usefulness of their collected dataset, where the fine-tuned models shows better visual CoT capacity or even get interleaved visual-text reasoning ability from scratch.

**Strengths:**

1. The dataset focus on an important problem: how to collect visual chain-of-thought (CoT) data. Their collected dataset provide valuable resources for training better visual reasoning models.
2. The dataset is large-scale and diverse. It contains 182k traces across 18 domains.
3. The authors also finetune two models to support the usefulness of the dataset.

**Weaknesses:**

1. In the data curation process, the images are first prepared and them the text reasoning traces are generated based on the images. It means that the reasoning steps must be pre-decided and cannot easily modified and extended. For these synthetic problems, it also means it heavily depends on the human prior knowledge (especially on some authors manual design?)

**Questions:**

1. I am not sure whether the analysis in section 4 can support the claim the visual CoT is important. In the experiments, the 1MT and 2MT provides additional finished reasoning steps for the models. It should significantly reduce the difficulty of the reasoning tasks. So the performance gain may come from the provided finished reasoning steps, instead of the visual information itself. I guess it should be compared to a baseline where the test-reasoning steps are also provided as text-only CoT.
2. I believe a static dataset is useful but not enough because the VLM or LLM with both visual understanding and generation ability are rapidly developing. In my opinion, a better way is to collect data in a dynamic way. For example, the model can generate useful visual CoT by itself and then collect them and finetune on them in an iterative way. Do you believe your data collection method can be extended to such dynamic way?
3. How to create the visual CoT is an important problem. I believe the images in your section 3 should also contain some information that how to create them. The figure 5 is not directly related to the specific tasks.
4. A minor suggestion: There are too many related works in section 2 and it is hard to align them to the table 1 because in your table 1, the dataset use their abbreviated name while in section 2, you use the authors' name. I believe it is better to also provide the dataset name in section 2 for better alignment. And I also wonder whether all mentioned datasets are included in Table 1? If not, I suggest to provide a more complete table in the appendix.

---

> ### Author Response · Authors · 2025-11-22
>
> Thank you for your thoughtful feedback, as well as the recognition of the breadth of visual reasoning tasks and quality of reasoning traces in our dataset that fills a gap for collecting visual CoT. We address your comments in detail below. We have also incorporated edits, which are reflected in the updated version of the PDF.
>
> ### 1. For these synthetic problems, it also means it heavily depends on the human prior knowledge (especially on some authors manual design?)
>
> This is an excellent question. We agree that this limitation applies broadly to SFT datasets. Even for most text CoT, it inevitably contains a fixed set of reasoning categories. Generating highly diverse visual CoT is non-trivial. Prior work typically focuses on a single domain and includes only a small number of visual CoT types. While we have made substantial efforts to broaden domain coverage and improve diversity, our approach is still grounded in human prior knowledge as you mentioned. We expect that true emergent behaviors would require methods such as RL (as mentioned in Future Work).
>
> However, we still believe that Zebra-CoT serves as the necessary "Cold Start" dataset. To train a model to reason with visual aids, it must first be able to inherently generate visual and text cot when solving problems, which no model can do that as far as we know. Once a model is fine-tuned on Zebra-CoT, such as Bagel, it forms the necessary initialization for RL training where the model can explore diverse reasoning paths.
>
> ### 2. Text-only CoT.
>
> We ran a text only CoT experiment as you described, where V is visual CoT and T is text only CoT
>
> | Provider | Q (%) | V-FT (%) | T-FT (%) | V-ST (%) | T-ST (%) |
> |----------|-------|----------|----------|----------|----------|
> | claude   | 27.61 | 45.00    | 39.57    | 58.70    | 38.48    |
> | gemini   | 24.93 | 42.26    | 17.88    | 48.29    | 21.01    |
> | gpt      | 41.98 | 56.70    | 46.15    | 63.08    | 54.29    |
>
> Notably, visual CoT constantly beats text only, and sometimes it’s even hurtful (for gemini mostly) to have those text-only CoT. This is because the text CoT is highly complementary to the visual reasoning data, and when only providing the text CoT, sometimes the model just asks for the visual aids because it noticed that it was missing. In general, we can see the that with visual CoT with text and images outperforms the same reasoning traces with only text CoT.
>
> ### 3. "In my opinion, a better way is to collect data in a dynamic way. For example, the model can generate useful visual CoT by itself and then collect them and finetune on them in an iterative way."
>
> Great question. It’s great that a model can generate visual CoT and then iteratively finetune, which would be considered as some sort of iterative self distillation and synthetic data generation. However, current multimodal generative models are still far from being capable of generating high quality visual CoT like text model synthetic data.
>
> One promising direction, which aligns with your suggestion, is to build an agentic pipeline where the model can call external tools (e.g., Python, Matplotlib, rendering engines) to generate visual aids during problem solving. This is exactly what we explored in our algorithmic agent setup: agents can programmatically produce graphs and figures that serve as high-quality visual CoT. We believe this tool-augmented, agent-based approach is a practical path toward dynamic data collection, which is mostly explored in the work Visual Sketchpad [2].
>
> ### 4. How to create the visual CoT is an important problem. I believe the images in your section 3 should also contain some information that how to create them. The figure 5 is not directly related to the specific tasks.
>
> Thank you for your question. Figure 5 is a high level schematic diagram for illustrating our pipeline. We have detailed the creation pipelines for every sub-category in Appendix A.3-A.6 and the prompt templates in F. We also included some high level clarification points in **general response 1**.
>
> Based on your suggestion, we edited the paper based on your suggestions. Now we explicitly redirect readers to not only Appendix A.2 and Figure 5 but also Appendix A.3-A.6 for sub-category details.
>
> ### 5. "A minor suggestion: There are too many related works in section 2 and it is hard to align them to the table 1 because in your table 1...".
>
> Thank you for this suggestion. We edited the paper update Section 2 and Table 1 to ensure consistent citation naming conventions as you suggested.
>
> [2] Hu, Yushi, et al. "Visual sketchpad: Sketching as a visual chain of thought for multimodal language models." Advances in Neural Information Processing Systems 37 (2024): 139348-139379.
>
> Thank you again for your valuable feedback. We have made paper edits in response to your feedback. We believe our novel dataset provides insight for future research in visual CoT, and we would greatly appreciate it if you would consider increasing your score accordingly.

---

> > ### Comment · Reviewer_UEnu · 2025-11-24
> >
> > Some of my questions have been solved. But my main question is about the dynamic generation or generalizations of dataset   could be considered as a future work, so I keep my score as boardline accept.

---

### Official Review · Reviewer_VPmE · 2025-10-26

**Soundness:** 3
**Presentation:** 3
**Contribution:** 3
**Rating:** 6
**Confidence:** 4

**Summary:**

This paper introduces Zebra-CoT, a large-scale, multi-domain interleaved text-image reasoning dataset designed to address the core issue of scarce high-quality visual chain-of-thought training data. The dataset covers four major categories of natural tasks particularly suited for visual reasoning: scientific problems, 2D/3D visual reasoning, and logical strategy tasks. Models fine-tuned on this dataset achieved significant performance improvements.

**Strengths:**

Scale and Diversity: The dataset is extensive (over 180K reasoning traces) and covers a wide range of domains (18 domains, 50+ distinct tasks), providing rich and comprehensive learning material for visual reasoning.

Relevance and Naturalness: The dataset is specifically constructed for task types where visual aids naturally enhance reasoning, ensuring high relevance and learning efficiency.

Empirically Validated Effectiveness: Experiments on multiple mainstream models demonstrate that the dataset significantly improves visual reasoning accuracy and the ability to generate high-quality visual reasoning chains.

The main contribution is the training dataset for the community.

**Weaknesses:**

1. After fine-tuning, the model demonstrates improved overall performance. However, to pinpoint which specific subcategories contribute to its gains on benchmarks like MathVision, MathVista, VisuLogic, EMMA, MMVP, Blink, and Vstar, a more detailed analysis is required due to the limited baseline comparisons currently available in these areas.

2. A detailed comparative analysis with prior benchmarks should be provided in the main paper, quantitatively examining aspects such as dataset scale and evaluation dimensions.

**Questions:**

N/A

---

> ### Author Response · Authors · 2025-11-21
>
> Thank you for your thoughtful feedback and recognition of the scale and diversity of visual reasoning tasks and quality of reasoning traces in our visual CoT training dataset. We address your comments in detail below. We have also incorporated edits to the paper based on your suggestions, which are reflected in the updated version of the PDF.
>
> ### 1. After fine-tuning, the model demonstrates improved overall performance. However, to pinpoint which specific subcategories contribute to its gains on benchmarks like MathVision, MathVista, VisuLogic, EMMA, MMVP, Blink, and Vstar, a more detailed analysis is required due to the limited baseline comparisons currently available in these areas.
>
> We thank you for this question. We included a detailed performance breakdown of the model in appendix C that gives some fine-grained understanding of downstream performance on sub domains. For the scaffolding experiment, we also show that some tasks benefit more than others, such as maze. For a comprehensive analysis of which data contributed to which performance, this is essentially a data mixture problem. Given the compute budget we have, we leave this as an interesting open problem and future work.
>
> ### 2. A detailed comparative analysis with prior benchmarks should be provided in the main paper, quantitatively examining aspects such as dataset scale and evaluation dimensions.
>
> Thank you for your question. We would like to first note that we are a training dataset rather than a benchmark. As noted in the **second point in our general response**, Zebra-CoT is the first dataset for multi step, interleaved text visual CoT dataset. The closest is Visual-CoT [1], which focuses on bounding boxes for visual search with single step visual CoT. Our dataset covers a much broader spectrum (scientific, 3D, logic). Comparing our generative visual reasoning model against a model fine-tuned on text-only CoT or detection-based CoT would be an apples-to-oranges comparison, as those models cannot generate the sketches/diagrams that are the core contribution of our work. Table 1 in the paper highlights these structural differences.
>
> Thank you again for your valuable feedback.  We have made paper edits in response to your feedback, which we think have improved our paper.  We would greatly appreciate it if you would consider increasing your score accordingly.  We believe our novel dataset of interleaved text and images provides insight and serves as a strong training dataset for future research in multimodal generative model training. We hope it will spark further intellectual exploration and discussion within the ICLR community. Please let us know if there are any remaining questions we can address.
>
> [1] Shao, Hao, et al. "Visual cot: Advancing multi-modal language models with a comprehensive dataset and benchmark for chain-of-thought reasoning." Advances in Neural Information Processing Systems 37 (2024): 8612-8642.

---

> ### Author Response · Authors · 2025-11-26
>
> Thanks again for your feedback. We made a strong effort to address all your points, including substantial edits to our draft, and we would greatly appreciate it if you would consider increasing your score accordingly. Do you have any other questions?

---

### Official Review · Reviewer_8hSM · 2025-10-30

**Soundness:** 3
**Presentation:** 4
**Contribution:** 4
**Rating:** 8
**Confidence:** 2

**Summary:**

The paper introduces ZEBRA-CoT, a large-scale dataset of 182K interleaved text–image reasoning traces designed to teach models to perform Visual Chain-of-Thought (visual CoT) reasoning. It contains 18 domains across four categories, e.g., scientific reasoning, 2D and 3D visual reasoning, and visual logic games.
Experiments show that frontier multimodal models (including GPT-5, Claude-4 Sonnet, Gemini 2.5 Pro) perform poorly on ZEBRA-CoT , but their performance improves when given the first one or two multimodal reasoning steps as scaffolding.
Overall, ZEBRA-CoT provides the first established dataset enabling models to think with images, representing a significant step toward interpretable and general multimodal reasoning.

**Strengths:**

- The paper addresses an important research question to enable models to reason visually rather than purely through text. It proposes a novel curated dataset (ZEBRA-CoT) that directly facilitates this capability beyond simple captioning or visual QA.

- The four categories (scientific, 2D, 3D, strategic) demonstrate careful task design and strong diversity, enabling generalization across reasoning types rather than a single narrow task.

-  The scaffolding experiments convincingly show that models benefit from intermediate multimodal reasoning cues, quantifying the value of explicit visual CoT supervision.

- Fine-tuning both Anole-7B and Bagel-7B demonstrates the dataset’s practicality and shows real gains on external benchmarks such as MathVista, VisuLogic, and MMVP.

- Establishing visual CoT as a training paradigm parallels the importance of textual CoT in LLM reasoning, with potential long-term implications for interpretable and grounded AI.

**Weaknesses:**

- While diversity is emphasized, there is little quantitative or human evaluation of logical coherence, correctness, or visual–text alignment within reasoning traces. Including human consistency checks or inter-annotator agreement would strengthen credibility.

- The paper does not isolate how different types of visual reasoning steps (e.g., sketches vs. crops vs. bounding boxes) contribute to performance. A finer-grained analysis would clarify which modalities drive improvement.

- Many examples rely on synthetic visual reasoning (mazes, jigsaws, diagrams).  Demonstrations on real-world, noisy visual data would better illustrate applicability.

- The limitation of the proposed dataset is not explicitly discussed. Elaborated discussions on the limitations or possible improvements would enhance the clarity of the paper.

**Questions:**

- How is visual coherence between text and generated sketches quantitatively verified?
- How does model scale affect the emergence of visual CoT?
- What are the major limitations of the proposed dataset? Is it ensured that it does not contain any noise or mislabeled examples?

---

> ### Author Response · Authors · 2025-11-21
>
> Thank you for your highly encouraging feedback and recognition of the breadth and diversity of visual reasoning tasks and quality of multi step interleaved text and visual reasoning traces in our dataset, as well as the importance of our downstream experiments. We address your comments in detail below. We have also incorporated edits to the paper based on your suggestions, which are reflected in the updated version of the PDF.
>
> ### 1. “While diversity is emphasized, there is little quantitative or human evaluation of logical coherence, correctness, or visual–text alignment within reasoning traces. Including human consistency checks or inter-annotator agreement would strengthen credibility.” “How is visual coherence between text and generated sketches quantitatively verified?” “ What are the major limitations of the proposed dataset? Is it ensured that it does not contain any noise or mislabeled examples?”
>
> Thank you for this question. We clarified about our pipeline creation in the **first point of the general response**. Regarding the logical coherence and correctness of the dataset, our pipeline is designed to ensure correctness by construction as we mentioned in the general response as well. You can also see Appendix A.2 (high level pipeline), A.3 - A.6 (sub category details), F (prompt templates) for more details.
>
> For real world data, the raw are from highly reliable sources like text book, and for synthetic data, all reasoning templates are python generated which are correct by design as well. For all data, we NEVER asked VLMs to generate images, or generate reasoning traces from scratch. For both, VLMs are only used for parsing or diversifying the text traces, while keeping all the key reasoning steps intact. Additionally, we manually verified some of the traces and used programmatic filtering such as regex to ensure those are high quality. As mentioned in A.2, “We further filter out invalid cases such as multiple image placeholders referring to the same image and unreferenced image placeholders to make sure that the data can be automatically parsed into a training dataset.”
>
> ### 2. The paper does not isolate how different types of visual reasoning steps (e.g., sketches vs. crops vs. bounding boxes) contribute to performance. A finer-grained analysis would clarify which modalities drive improvement.
>
> We thank you for this question. We included a detailed performance breakdown of the model in appendix C that gives some fine-grained understanding of downstream performance on sub domains. For the scaffolding experiment, we also show that some tasks benefit more than others, such as maze. For a comprehensive analysis of which data contributed to which performance, this is essentially a data mixture problem. Given the compute budget we have, we leave this as an interesting open problem and future work.
>
> ### 3. Many examples rely on synthetic visual reasoning (mazes, jigsaws, diagrams). Demonstrations on real-world, noisy visual data would better illustrate applicability.
>
> Thank you for your question. We do have a lot of real world data, such as geometry, physics, chemistry, visual search, and robotic planning data. For those data, we took the original reasoning traces, such as illustrations in the text book, and asked VLMs to parse into clean visual cot data that are logically coherent and refers back to visual aids to strengthen interleaved reasoning between the two modalities. For robotic planning, the original instructions are high quality and diverse enough so we leave it as it is.
>
> ### 4. The limitation of the proposed dataset is not explicitly discussed. Elaborated discussions on the limitations or possible improvements would enhance the clarity of the paper.
>
> Thank you for this suggestion. We agree that while we substantially broadens the diversity of visual reasoning traces compared to prior work, it still has several limitations. One limitation is that the set of reasoning traces is still confined with a fixed number of categories. In the future, we would like to diversify this more by including tasks such as 3D tasks and stronger temporal reasoning like video reasoning data. Another limitation is that current reasoning traces are constructed based on human prior, i.e. we create the dataset by mimicking how humans would think in visual spaces. VLMs, however, might tackle multimodal thinking in a different way that is drastically different from human priors. One approach for solving this limitation to train our fine-tuned model with RL and see if we can have emergent behaviors. We leave this as future work.
>
> ### 5. How does model scale affect the emergence of visual CoT?
>
> Thank you for this question. As far as we know, no model can think with visual CoT explicitly, even SoTA models like GPT-5. For studying the emergence of visual CoT, we leave it as an open question for future work.
>
> Thank you again for your valuable and highly positive feedback! Let us know if you have any further questions.

---

> ### Author Response · Authors · 2025-11-26
>
> Thanks again for your feedback. We made a strong effort to address all your points, including substantial edits to our draft, and we would greatly appreciate it if you would consider increasing your score accordingly. Do you have any other questions?

---

### Official Review · Reviewer_hQr7 · 2025-11-04

**Soundness:** 4
**Presentation:** 4
**Contribution:** 4
**Rating:** 6
**Confidence:** 3

**Summary:**

The paper proposes a new vision-language reasoning dataset Zebra. It contains 182k reasoning chains featured with interleaved visual and textual reasoning steps. Experiments on two VLM backbones show substaintial performance gains after fine-tuning on the proposed Zebra dataset.

**Strengths:**

+ It proposes a high-quality dataset featured with interleaved visual reasoning chains on multiple domains, fill in the gap of previous visual cot datasets
+ The pilot scalffording experiments and the fine-tuning experiment shows that the dataset could improve downstream visual reasoning performance.
+ The paper is well written and easy to follow.

**Weaknesses:**

+ Some details of the dataset curation process is missing. From Appendix A.2 we know that the dataset contains both real-world data and synthetic data, but the author does not reveal the detailed source of the real-world data or the specific process of how the synthetic data is generatd. This is important because we don't know if the evaluation dataset like MathVista or MathVision is contaminated or not. Testing the high-similarity overlap between the proposed dataset and the evaluation benchmark would make the performance gains more convincing.

+ I would suggest that In section 5 the fine-tuning experiment to include fine-tuning on previous CoT dataset for a more direct comaprision of your data quality with previous ones.

**Questions:**

See the weaknesses above.

---

> ### Author Response · Authors · 2025-11-21
>
> Thank you for your thoughtful feedback and recognition of the breadth of visual reasoning tasks and quality of reasoning traces in our dataset that fills a gap for visual CoT. We address your comments in detail below. We have also incorporated edits to the paper based on your suggestions, which are reflected in the updated version of the PDF.
>
> ### 1. “Some details of the dataset curation process is missing. From Appendix A.2 we know that the dataset contains both real-world data and synthetic data, but the author does not reveal the detailed source of the real-world data or the specific process of how the synthetic data is generatd. This is important because we don't know if the evaluation dataset like MathVista or MathVision is contaminated or not. Testing the high-similarity overlap between the proposed dataset and the evaluation benchmark would make the performance gains more convincing.”
>
> Thank you for raising concerns about data transparency. We documented the source and generation process for all subdomains in Appendix A.2 through A.6. We also clarified this in the **first point of our general response** above. We agree that ensuring contamination is a critical point. To minimize potential overlap with MathVista and MathVision, we carefully selected our data sources. For example, the geometry subset is generated directly from the original MATH dataset, and for multi-hop evaluation, we render entirely new datasets in Unity with different styles and scenes rather than relying on CLEVR-style data used in MathVista or MathVision.
>
> Regarding potential high-similarity overlap between our dataset and the evaluation benchmarks, we acknowledge that there might be some similar problem categories, which is unavoidable. However, MathVista and MathVision contain many categories that do not appear in our training data. Most importantly, as shown in Appendix B, our model still generalizes and produces novel emergent visual aids that were not present in the training set.
>
> ### 2. "I would suggest that In section 5 the fine-tuning experiment to include fine-tuning on previous CoT dataset for a more direct comaprision of your data quality with previous ones."
>
> Thank you for your question. As noted in the **second point in our general response**, Zebra-CoT is the first dataset for multi step, interleaved text visual CoT dataset. The closest is Visual-CoT [1], which focuses on bounding boxes for visual search with single step visual CoT. Our dataset covers a much broader spectrum (scientific, 3D, logic). Comparing our generative visual reasoning model against a model fine-tuned on text-only CoT or detection-based CoT would be an apples-to-oranges comparison, as those models cannot generate the sketches/diagrams that are the core contribution of our work. Table 1 in the paper highlights these structural differences.
>
> Thank you again for your valuable feedback.  We have made paper edits in response to your feedback, which we think have improved our paper.  We would greatly appreciate it if you would consider increasing your score accordingly.  We believe our novel dataset of interleaved text and images provides insight and serves as a strong training dataset for future research in multimodal generative model training. We hope it will spark further intellectual exploration and discussion within the ICLR community. Please let us know if there are any remaining questions we can address.
>
> [1] Shao, Hao, et al. "Visual cot: Advancing multi-modal language models with a comprehensive dataset and benchmark for chain-of-thought reasoning." Advances in Neural Information Processing Systems 37 (2024): 8612-8642.

---

> ### Author Response · Authors · 2025-11-26
>
> Thanks again for your feedback. We made a strong effort to address all your points, including substantial edits to our draft, and we would greatly appreciate it if you would consider increasing your score accordingly. Do you have any other questions?

---

### Author Response · Authors · 2025-11-21

We thank all reviewers for their thoughtful feedback and for their highly positive evaluations of our work. We are encouraged that reviewers found Zebra-CoT to be a high-quality, diverse, and novel dataset that fills an important gap in multimodal reasoning. We also appreciate the recognition of our core contribution: curating a visual CoT dataset where visual aids naturally enhance and augment the reasoning trajectories, enabling models to reason in both text and visual spaces. Finally, we thank the reviewers for highlighting that our fine-tuning and scaffolding experiments clearly demonstrate the usefulness and impact of the dataset.

Below, we address common questions regarding data curation and baselines, followed by specific responses to each reviewer.

## Clarification on dataset curation details (Reviewers hQr7, 8hSM, UEnu):

There is a shared question regarding the details of our data. Since we have a wide range of domains (18 domains, 50+ distinct tasks), we included all the details in Appendix A.2 (high level pipeline), A.3 - A.6 (sub category details), F (prompt templates).

Specifically, we include both real world data and synthetically generated program verifiable data, and by our careful design choices, we ensure that the data are correct and logically coherent.

**Real world data**: For domains like science, we source high-quality problems from established repositories (e.g., MATH, OpenStax Physics, USPTO-50K). For those stem reasoning data, we do not just use the existing text as they are in raw format and lack logical connections that will stimulate models to generate those visual aids. We use VLMs to parse the current traces into high quality training data. For some other real world domains such as robotic planning and embodied reasoning, we directly use the text we sourced as they are high quality and diverse enough.

**Synthetic and programmatic data**: For domains like Mazes, Chess, and Algorithms, we use programmatic engines (e.g., Stockfish, Python graph libraries, and LLM Agent) to generate dataset based on our constructed templates. Those data are correct by construction, as we generated the reasoning traces and answers based on verifiable programs. For those, the drawback is the lack of diversity in text traces. We therefore call frontier VLMs to enhance the diversity of those text traces. The specific prompt templates can be found in Appendix F.

An important thing is that for all data, we **NEVER** asked VLMs to generate images, or generate reasoning traces from scratch. For both, VLMs are only used for parsing or diversifying the text traces, while keeping all the key reasoning steps intact. Additionally, we manually verified some of the traces and used programmatic filtering such as regex to ensure those are high quality. As mentioned in A.2, “We further filter out invalid cases such as multiple image placeholders referring to the same image and unreferenced image placeholders to make sure that the data can be automatically parsed into a training dataset.”

## Comparison with Prior visual CoT Datasets (Reviewers hQr7, VPmE)

Reviewers suggested comparing against previous visual CoT datasets quantitatively. We respectfully emphasize that there are only a few equivalent that exist that allows us for a meaningful quantitative comparison. We include the detailed qualitative analysis in section 2 and Table 1, and we offer a high level comparison below with key distinctions:

**Multimodal Reasoning with text-only CoT**: GQA, ScienceQA, R1-Onevision are datasets with multimodal input, and text-only CoT, while Zebra-CoT focus on interleaved text and visual CoT.

**Other visual CoT**: Visual CoT focuses almost exclusively on single step visual search/bounding boxes, MM-PhyQA focuses on physics data, and it’s not open-sourced. CoT VLA is interleaved image and action, without text, and only for robotics.

**Interleaved text-image data (e.g., MINT-1T, OBELICS, OmniCorpus)**: These are web-scraped and lack the step-by-step logical coherence of a reasoning chain. Zebra-CoT is the first large-scale dataset designed specifically for generating interleaved visual reasoning.

In conclusion, we believe that we are the first to create a diverse set of multi step interleaved text-visual CoT dataset that spans multiple category and sub domains.

---

### Meta-Review · Area_Chair_QBG9 · 2026-01-05

**Summary:**

This paper introduces Zebra-CoT, a large-scale dataset of interleaved text–image reasoning traces intended to train multimodal models to perform visual chain-of-thought. The dataset is large (182k traces) and broad (18 domains, 50+ tasks), and the paper includes fine-tuning and scaffolding experiments that show consistent downstream gains. Reviewers are broadly positive and view this as a useful dataset contribution for the community.

**Reviewer Concerns:**

Main points raised by reviewers were mostly about dataset rigor and analysis rather than the core idea:
- more transparency on data sources/curation and stronger contamination checks against evaluation benchmarks
- request for additional comparisons to prior visual CoT datasets
- limited quantitative/human evaluation of trace correctness and visual–text alignment; request for more explicit limitations and noise discussion
- request for ablations on which subcategories/types of visual steps (sketches vs crops vs boxes) drive the gains
- concern that scaffolding gains might be partly from providing intermediate reasoning steps, asking for a text-only CoT control

The AC finds the rebuttal to be responsive, and the authors appear to be successfully addressing most of the practical concerns. Authors clarify the curation pipeline, distinguish real-world vs synthetic sources, and explain how VLMs are used. They also address contamination concerns directly, discussing source and dataset rendering choices. They provide the requested text-only CoT baseline, showing that visual CoT beats text-only CoT in their setup, directly addressing the key confound concern. Some requests (full mixture attribution of which subdomains drive which benchmark gains, deeper human evaluation) are acknowledged as important but left as future work due to compute and scale.

**Reviewer Scores:**

Pre-rebuttal scores are all above the threshold, with one reviewer clearly positive and others at borderline accept. Post-rebuttal, large shifts are unlikely, but the rebuttal reduces the main methodological doubts.
- reviewer 8hSM: 8, likely unchanged
- reviewer hQr7: 6, likely unchanged or slightly higher
- reviewer VPmE: 6, likely unchanged or slightly higher
- reviewer UEnu: 6, unchanged (explicitly keeps borderline accept)

---

### Decision · Program_Chairs · 2026-01-26

Accept (Poster)